# Hidden Conformal Symmetries from Killing Towers with an Application to Large-D/CFT

Cynthia Keeler,[1] Victoria Martin,[2] and Alankrita Priya[1]

[1]*Department of Physics, Arizona State University, Tempe, AZ, 85287, USA*

[2]*University of Iceland, Science Institute, Dunhaga 5, IS-107, Reykjavik, Iceland*

*E-mail:* keelerc@asu.edu, vlmartin@hi.is, apriya@asu.edu

ABSTRACT: We generalize the notion of hidden conformal symmetry in Kerr/CFT to Kerr-(A)dS black holes in arbitrary dimensions. We build the $SL(2, R)$ generators directly from the Killing tower, whose Killing tensors and Killing vectors enforce the separability of the equations of motion. Our construction amounts to an explicit relationship between hidden conformal symmetries and Killing tensors: we use the Killing tower to build a novel tensor equation connecting the $SL(2, R)$ Casimir with the radial Klein-Gordon operator. For asymptotically flat black holes in four and five dimensions we recover previously known results that were obtained using the "near-region" limit and the monodromy method. We then perform a monodromy evaluation of the Klein-Gordon scalar wave equation for all Kerr-(A)dS black holes, finding explicit forms for the zero mode symmetry generators. We also extend this analysis to the large-dimensional Schwarzschild black hole as a step towards buliding a Large-D/CFT correspondence.

## 1 Introduction

The powerful insight that the physics of a gravitational system can be described via a lower-dimensional conformal field theory (CFT) has evolved much in the past 20 years. Beyond the celebrated AdS/CFT correspondence [1], a CFT description of an extremal Kerr black hole in flat space was achieved by [2] in a near-horizon limit. This Kerr/CFT correspondence not only provided a flat space example of a gauge/gravity duality, but it also helped provide a microscopic accounting of the Bekenstein-Hawking entropy of the extremal Kerr black hole,

assuming the validity of a Cardy formula. Since then, Kerr/CFT has been extended to higher-dimensional extremal black holes [3–7]. It was then of immediate interest to determine the extent to which such conformal structure exists away from extremality.

The possibility of a CFT description of the Kerr black hole away from extremality was first studied by [8]. There they considered the dynamics of a massless scalar field on a generic Kerr background. They found that conformal symmetry was preserved away from extremality, provided that a near-region limit was taken in the Klein-Gordon equation (as opposed to the metric)[1]. Since the conformal symmetry of the non-extremal Kerr solution is only discernable by studying the dynamics rather than the background geometry alone, this feature is referred to as a *hidden conformal symmetry*. A corresponding near-region analysis has now been applied to many non-extremal black hole systems, and hidden conformal symmetry has thus been shown to be a generic feature of black hole backgrounds [10–12]. The physical connection between this near-region approach of finding hidden conformal symmetry and black hole soft hair [13] is treated in [9, 14, 15].

There is a second method of diagnosing hidden conformal symmetry in black hole backgrounds, known as the monodromy method [16–19]. While the spirit of the monodromy and near-region methods is the same (studying the near-horizon behavior of the wave equation), the former does not require taking a near-region/soft hair limit by hand. The hidden conformal symmetry generators for Kerr and the $5D$ Myers-Perry black hole were found using the monodromy method in [18], reproducing the near-region results of [8, 10]. In this paper we will use both the near-region and monodromy approaches extensively, and so we review them in Section 2.

Intriguingly, the concept of hidden symmetry has played a powerful role in a seemingly unrelated area of physics: separability and integrablilty of equations of motion [20, 21]. That is, separability of the Klein-Gordon equation and total integrability of geodesic motion is *guaranteed* if the spacetime admits a principal tensor[2] which generates a tower of Killing tensors and Killing vectors. A Killing tensor is a higher rank object that satisfies Killing's tensor equation $\nabla_{(\mu} k_{\alpha\beta)} = 0$. Remarkably, it was shown by [20, 22] that the most general Kerr-NUT-(A)dS spacetime in any number of dimensions admits enough Killing tensors that all equations of motion are separable. Unlike Killing vectors $l_{(i)}$, which generate *explicit symmetries* of the metric (isometries) with conserved quantities linear in momentum $l^a p_a$, Killing tensors generate *hidden symmetries* of the dynamics, with conserved quantities that are higher order[3] in momenta $k^{ab} p_a p_b$. For an extensive review on separability, hidden symmetries and Killing tensors, see [21].

---

[1]The authors of [8] refer to this as a "near-region limit" in order to distinguish it from the near-horizon limit taken in the metric, as in Kerr/CFT. In [9], the authors describe this as a near-horizon limit in the dynamics. As we will discuss in Section 2, the near-region limit is defined by $\omega M << 1$ and $\omega r << 1$, where $\omega$ is the eigenvalue of $i\partial_t$ and $M$ is the black hole mass.

[2]This is more correctly called a non-degenerate closed conformal Killing-Yano 2-form.

[3]In general there exist Killing-Yano tensors of the form $f^{c_1 \dots c_{D-2j-1}}$; the Killing tensors we use here are built from the square of the $j$th Killing-Yano: $k_{(j)}^{ab} \propto f^{(j)}_{c_1 \dots c_{D-2j-1}} f^{(j)bc_1 \dots c_{D-2j-1}}$. For a full list of objects in the Killing tower, see section 5 of [21].

The fact that the same black hole spacetimes that exhibit hidden conformal symmetry in a near-region limit also admit a Killing-Yano tensor that ensures separability has already been observed by [23, 24]. In this paper, we make these connections precise. We use the Killing tower of [21] to construct a tensor equation for the quadratic Casimir $\mathcal{H}^2$ of the hidden conformal symmetry generators for Kerr-(A)dS spacetimes. We construct the general form of this tensor equation for all dimensions, and compute explicit expressions in $4D$ and $5D$ with $\Lambda = 0$. From the monodromy point of view we can go further, and build the monodromy parameters $\alpha_\pm$ (to be defined in Section 2.1.2) in general spacetime dimensions from the Killing tower, recovering in particular results previously obtained for Kerr and the $5D$ Myers-Perry black hole [18] and Kerr-AdS [15]. This connection is of particular interest, as it may point to a thermodynamic interpretation of the Killing tower.

In addition, it is interesting to consider the large dimension (large $D$) limit of black hole spacetimes ([25–34] or for a recent full review, see [35]), as calculations in this limit are greatly simplified without compromising the near-horizon physics. It is particularly interesting to study hidden conformal symmetry in the large $D$ limit, as this could point to a Large-$D$/CFT correspondence.[4] In the present work, we attempt to construct a Killing tensor equation for the quadratic Casimir in the large $D$ limit, using two different approaches. First, we take the large $D$ limit in the metric (as is standard practice), and then we take the large $D$ limit directly in the wave equation. We discuss the results of the former approach, and the complications of the latter.

This paper is structured as follows. In Section 2 we review the topics that are central to this work. Section 2.1 focuses on the near-region and monodromy methods for finding the hidden conformal symmetry generators, and Section 2.2 reviews the Killing tower construction and separability of the wave equation. In Section 3 we generalize the hidden conformal symmetry generators of [8] to general dimension, which is necessary to construct a tensor equation for the quadratic Casimir. We build the full tensor equation for four and five dimensions, and highlight the difficulty that arises when considering general dimension. In Section 4 we construct the monodromy parameters $\alpha_\pm$ from the Killing tower for the general $D$ Kerr-(A)dS spacetime, allowing us to compute the hidden conformal symmetry generators from the Killing tower as well. In Section 5 we match the Casimir and perform a monodromy analysis for large $D$ Schwarzschild-Tangherlini black holes. A discussion of our results and future work is given in Section 6. In Appendix A we set our notation and write down the metrics that we use. Appendix B gives a physical discussion of the monodromy basis introduced in Section 2.1.2.

---

[4]The large $D$ quadratic Casimir of the hidden symmetry generators of charged AdS black holes was found in [36]; for the charged case, they find the temperatures but do not build the $H$ generators or the conformal coordinates explicitly.

## 2  Review: Hidden Conformal Symmetry, Killing Tensors and Separability

Here we review two separate manifestations of hidden symmetry. The first is the hidden conformal symmetries obtained by analyzing the wave equation in [8, 18]. The second is the hidden symmetries as generated from Killing tensors in [21].

### 2.1  Hidden Conformal Symmetry in Kerr/CFT

With the establishment of the Kerr/CFT correspondence [2], the next question that arises is whether conformal symmetry also exists away from extremality. In this section we review two programs that obtain hidden conformal symmetry generators in the Kerr background and the five-dimensional Myers-Perry black hole.

#### 2.1.1  Near-region limit

The first progress in uncovering hidden conformal symmetry away from extremality was accomplished in [8]. The authors considered the Klein-Gordon equation for a massless scalar field $\Phi$ in a Kerr background. This equation famously separates under the ansatz $\Phi = R(r)S(\theta)e^{i(m\phi-\omega t)}$, and for convenience the radial equation is reproduced below

$$\left(\partial_r(\Delta\partial_r) + \frac{(2Mr_+\omega - am)^2}{(r-r_+)(r_+ - r_-)} - \frac{(2Mr_-\omega - am)^2}{(r-r_-)(r_+ - r_-)} + (r^2 + 2M(r+2M))\omega^2\right)R(r) = KR(r),$$
(2.1)

where $\Delta = r^2 + a^2 - 2Mr = (r-r_+)(r-r_-)$, $r_\pm = M \pm \sqrt{M^2 - a^2}$ are the inner and outer event horizons. $a = J/M$ is the black hole spin (with $M$ and $J$ the mass and angular momentum), and $K$ is a separation constant. In order to see hints of hidden conformal symmetry emerge, the authors of [8] took a near-region limit, defined by $\omega M << 1$ and $\omega r << 1$. In this limit, the final $\omega^2$ term in (2.1) vanishes:

$$\left(\partial_r(\Delta\partial_r) + \frac{(2Mr_+\omega - am)^2}{(r-r_+)(r_+ - r_-)} - \frac{(2Mr_-\omega - am)^2}{(r-r_-)(r_+ - r_-)}\right)R(r) = KR(r).$$
(2.2)

The resulting solutions are hypergeometric functions, which transform in representations of $SL(2,R)$. The authors of [9] refer to these low $\omega$ solutions as soft hair modes, and give an interpretation of the near-region limit as a near-horizon limit of phase space, rather than a near-horizon limit of spacetime: $\omega(r - r_+) << 1$.

To make the conformal symmetry more manifest, it is prudent to work in the *conformal coordinates* introduced in [8, 37] and subsequently utilized and adapted by [9, 10, 14–16, 18]

$$w^+ = \left(\frac{r - r_+}{r - r_-}\right)^{1/2} e^{2\pi T_R\phi}$$

$$w^- = \left(\frac{r - r_+}{r - r_-}\right)^{1/2} e^{2\pi T_L\phi - \frac{t}{2M}}$$
(2.3)

$$y = \left(\frac{r_+ - r_-}{r - r_-}\right)^{1/2} e^{\pi(T_L+T_R)\phi - \frac{t}{4M}},$$

where

$$T_R = \frac{r_+ - r_-}{4\pi a}, \qquad T_L = \frac{r_+ + r_-}{4\pi a}. \tag{2.4}$$

The surface $w^+ = 0$ is the past horizon, $w^- = 0$ defines the future horizon, and $w^\pm = 0$ is the bifurcation surface. The coordinates (2.3) are the Kerr analogue of the coordinate transformation which takes the BTZ black hole metric to the upper-half plane of $AdS_3$ in Poincaré coordinates [38]. That is, close to the bifurcation surface (to leading order in $w^\pm = 0$), the Kerr metric (A.1) becomes

$$ds^2 = \frac{4\rho_+^2}{y^2}dw^+ dw^- + \frac{16J^2 \sin^2\theta}{y^2 \rho_+^2}dy^2 + \rho_+^2 d\theta^2 + ..., \tag{2.5}$$

which is warped $AdS_3$ (for a given $\theta$ slice). Here $\rho_+^2 = r_+^2 + a^2\cos^2\theta$. Further motivation for these coordinates is discussed in Section 3.1.

The key point is now to construct the *locally defined* vector fields

$$\begin{aligned}
H_1 &= i\partial_+, & H_0 &= i(w^+\partial_+ + \frac{1}{2}y\partial_y), & H_{-1} &= i((w^+)^2\partial_+ + w^+ y\partial_y - y^2\partial_-), \\
\bar{H}_1 &= i\partial_-, & \bar{H}_0 &= i(w^-\partial_- + \frac{1}{2}y\partial_y), & \bar{H}_{-1} &= i((w^-)^2\partial_- + w^- y\partial_y - y^2\partial_+).
\end{aligned} \tag{2.6}$$

Each set of vector fields satisfies an $SL(2,R)$ algebra

$$[H_0, H_{\pm 1}] = \mp i H_{\pm 1}, \qquad [H_{-1}, H_1] = -2iH_0, \tag{2.7}$$

with quadratic Casimir

$$\begin{aligned}
\mathcal{H}^2 &= -H_0^2 + \frac{1}{2}\left(H_1 H_{-1} + H_{-1}H_1\right) \\
&= \frac{1}{4}(y^2\partial_y^2 - y\partial_y) + y^2\partial_+\partial_-,
\end{aligned} \tag{2.8}$$

(where analogous statements hold for the $\bar{H}$s). Upon taking the near-region limit, [8] showed that the quadratic Casimir (2.8) is precisely the radial Klein-Gordon operator (2.2), i.e.

$$\mathcal{H}^2\Phi = \bar{\mathcal{H}}^2\Phi = K\Phi. \tag{2.9}$$

This same approach was taken in [10] to find the analogous hidden conformal symmetry structure in the $5D$ Myers-Perry black hole. In five dimensions the separation ansatz for the wave equation solution is $\Phi = R(r)S(\theta)e^{i(m_1\phi_1 + m_2\phi_2 - \omega t)}$. It turns out that considering two specific types of waves (namely $m_1 = 0$ and $m_2 = 0$) gives rise to two unrelated CFTs, one in the $\phi_1$ sector and the other in the $\phi_2$ sector. To show this, the principal new step of [10] was to generalize the conformal coordinates to five dimensions. For example, in the $\phi_1$ sector we

have:

$$w^+ = \left(\frac{r^2 - r_+^2}{r^2 - r_-^2}\right)^{1/2} e^{2\pi T_R \phi_1 - 2K_R t}$$

$$w^- = \left(\frac{r^2 - r_+^2}{r^2 - r_-^2}\right)^{1/2} e^{2\pi T_L \phi_1 - 2K_L t} \tag{2.10}$$

$$y = \left(\frac{r_+^2 - r_-^2}{r^2 - r_-^2}\right)^{1/2} e^{\pi(T_L + T_R)\phi_1 - (K_L + K_R)t},$$

where

$$T_R = \frac{1}{\pi}\frac{\kappa_+(\kappa_- - \kappa_+)}{\Omega_R(\kappa_- - \kappa_+) - \Omega_L(\kappa_- + \kappa_+)}, \qquad T_R = \frac{1}{\pi}\frac{\kappa_+(\kappa_- + \kappa_+)}{\Omega_R(\kappa_- - \kappa_+) - \Omega_L(\kappa_- + \kappa_+)},$$

$$K_R = \frac{1}{\pi}\frac{\Omega_L \kappa_+ \kappa_-}{\Omega_R(\kappa_- - \kappa_+) - \Omega_L(\kappa_- + \kappa_+)}, \qquad K_R = \frac{1}{\pi}\frac{\Omega_R \kappa_+ \kappa_-}{\Omega_R(\kappa_- - \kappa_+) - \Omega_L(\kappa_- + \kappa_+)}, \tag{2.11}$$

$$\Omega_R = \Omega_{\phi_1} + \Omega_{\phi_2} = \frac{r_+(a_1 + a_2)(r_+ + r_-)}{(r_+^2 + a_1^2)(r_+^2 + a_2^2)},$$

$$\Omega_L = \Omega_{\phi_1} - \Omega_{\phi_2} = \frac{r_+(a_1 - a_2)(r_+ - r_-)}{(r_+^2 + a_1^2)(r_+^2 + a_2^2)}. \tag{2.12}$$

$\Omega_{\phi_1}$ and $\Omega_{\phi_2}$ are the angular velocities with respect to each angle, and $\kappa_\pm$ are the surface gravities at the inner and outer horizons. The barred versions are the same, except the right and left expressions are exchanged: $T_L \leftrightarrow T_R$ and $K_L \leftrightarrow K_R$. To obtain the expressions (2.10) - (2.12) for the $\phi_2$ sector, simply replace $\phi_1$ with $\phi_2$ and exchange $a_1 \leftrightarrow a_2$.

With this definition of the conformal coordinates, equations (2.6) - (2.9) hold for each independent sector ($m_1 = 0$ and $m_2 = 0$). In order to study the existence of hidden conformal symmetries in higher dimensions, our immediate objective is to generalize these conformal coordinates. We discuss this procedure in Section 3.1.

### 2.1.2 Monodromy method

There is a second method of probing hidden conformal symmetry which demonstrates that the $SL(2,R) \times SL(2,R)$ symmetry of the radial equation discovered in [8, 10] actually persists without taking the near-region limit in the Klein-Gordon equation. This *monodromy method* only requires examining the analytic structure of the full radial equation.

We begin by analyzing the regular singular points of a differential equation

$$R''(r) + P(r)R'(r) + Q(r)R(r) = 0. \tag{2.13}$$

This equation possesses a regular singular point $r_i$ if $P(r)$ or $Q(r)$ diverges as $r \to r_i$ but $(r - r_i)P(r)$ and $(r - r_i)^2 Q(r)$ remain finite as $r \to r_i$. Due to the branch cuts that form at these singular points, the solutions develop a monodromy when going around a singular point in the complex $r$ plane. To find the monodromy data around a given singular point $r_i$,

we consider a series solution for $R(r)$ near that singular point. The two solutions will be of the form

$$R_i^{out} = (r - r_i)^{i\alpha_i}(1 + \mathcal{O}(r - r_i)), \quad R_i^{in}(r) = (r - r_i)^{-i\alpha_i}(1 + \mathcal{O}(r - r_i)). \tag{2.14}$$

Here $\alpha_i$ is the monodromy parameter.

Let's again consider the example of a massless scalar field in the Kerr background. The radial equation (2.1) has two regular singular points at the horizons $r_+$ and $r_-$, and an irregular singular point at $r = \infty$[5]. In order to compute the monodromy parameter around $r = r_\pm$, we first express (2.1) in standard form:

$$(r - r_\pm)^2 R''(r) + (r - r_\pm)\lambda(r)R'(r) + \gamma(r)R(r) = 0 \tag{2.15}$$

where

$$\lambda(r) = (r - r_\pm)P(r) \tag{2.16}$$

and

$$\gamma(r) = (r - r_\pm)^2 Q(r). \tag{2.17}$$

We can solve the above differential equation around the singular point $r = r_\pm$ using the Frobenius method of series expansions. This gives the following indicial equation

$$\beta(\beta - 1) + \lambda_0\beta + \gamma_0 = 0, \tag{2.18}$$

where $\beta \equiv i\alpha$, $\lambda_0 \equiv \lambda(r_\pm)$ and $\gamma_0 \equiv \gamma(r_\pm)$. Solving the indicial equation gives the monodromy parameters $\alpha_\pm$ associated with the inner and outer horizons [16]

$$\alpha_\pm = \frac{\omega - \Omega_\pm k}{2\kappa_\pm}. \tag{2.19}$$

Expressions for angular velocities $\Omega_\pm \equiv \left.\frac{d\phi}{dt}\right|_{r=r_\pm}$ and surface gravities $\kappa_\pm$ are given in Appendix A.

As we will motivate in Section 3.1 and Appendix B, we employ the change of basis

$$\begin{aligned} \omega_L &= \alpha_+ - \alpha_- \\ \omega_R &= \alpha_+ + \alpha_-. \end{aligned} \tag{2.20}$$

The variables $t_L$ and $t_R$ conjugate to $\omega_L$ and $\omega_R$ are found by comparing the Fourier modes

$$e^{-i\omega_L t_L - i\omega_R t_R} = e^{-i\omega t + im\phi}. \tag{2.21}$$

---

[5] For the purposes of our discussion, we will not need to address the irregular singular point at infinity. Indeed, the monodromy information for this irregular singular point is already encoded in that of the two regular singular points via the relation $\mathcal{M}_+ \mathcal{M}_- \mathcal{M}_\infty = \mathbb{I}$, where $\mathcal{M}_i$ is the monodromy matrix around the $i$th singular point. The curious reader can consult for example [16] for further discussion.

Plugging in the explicit values of $\alpha_\pm$ in (2.19) and grouping the coefficients of $\omega$ and those of $m$, we find

$$t_R = 2\pi T_R \phi, \qquad t_L = \frac{1}{2M}t - 2\pi T_L \phi. \tag{2.22}$$

The new basis $(\omega_L, \omega_R)$ are eigenvalues of the operators $(i\partial_{t_L}, i\partial_{t_R})$. In the $(t, \phi)$ basis, these operators become the generators

$$H_0 = \frac{i}{2\pi T_R}\partial_\phi + 2iM\frac{T_L}{T_R}\partial_t, \qquad \bar{H}_0 = -2iM\partial_t \tag{2.23}$$

posited in [8]. Further, comparing the conformal coordinates in (2.3) and the left- and right-moving coordinates in (2.22) shows that $(t_L, t_R)$ fix the conformal coordinates themselves (up to an $r$-dependent scaling). Thus we are able to completely fix the exponential factors of the $(H, \bar{H})$ in (2.6) using the monodromy method. The analogous monodromy calculation was presented for the $5D$ Myers-Perry black hole in [16], and their results matched those of [10].

## 2.2  Hidden Symmetry, Killing Tensors and Separability

We now review the Killing vectors and Killing tensors responsible for the separability of wave equation in general Kerr-NUT-(A)dS black hole spacetimes. As reviewed in [21], spacetimes which possess a non-degenerate closed conformal Killing-Yano 2-form, also known as a principal tensor, must additionally have a full tower of Killing objects. As the authors of [20, 22, 39] showed, these spacetimes consequently have enough Killing tensors to separate all equations of motion.

For this paper, we will not need the details of the principal tensor or its entire associated Killing tower (see [21] for a thorough review of these topics). Instead, we will focus on the Killing vectors and Killing tensors of the Kerr-(A)dS geometries.

We begin by stating the metric for the Kerr-NUT-(A)dS geometry in canonical coordinates, first found in [40]:

$$ds^2 = \sum_{\mu=1}^{n}\left[\frac{U_\mu}{X_\mu}\,dx_\mu^2 + \frac{X_\mu}{U_\mu}\Big(\sum_{j=0}^{n-1}A_\mu^{(j)}d\psi_j\Big)^2\right] + \epsilon\frac{c}{A^{(n)}}\Big(\sum_{k=0}^{n}A^{(k)}d\psi_k\Big)^2. \tag{2.24}$$

Here $D = 2n+\epsilon$, so $\epsilon = 0$ for even $D$ and $\epsilon = 1$ for odd $D$, and $c$ is an arbitrary constant and is fixed by the transformation to Boyer-Lindquist-like coordinates. The radial direction is given by $x_n = ir$, while the $x_\mu$, $\mu = 1, \ldots, n$, correspond to longitudinal directions. The Killing directions $\psi_k$, where $k = 0, \ldots, n+\epsilon-1$, are related to the $\phi_\mu$ and $t$ of the Boyer-Lindquist-like coordinates (A.4) via

$$t = \psi_0 + \sum_{k=1}^{n+\epsilon-1}\mathcal{A}^{(k)}\psi_k, \quad \frac{\phi_\mu}{a_\mu} = \lambda\psi_0 - \sum_{k=1}^{n+\epsilon-1}(\mathcal{A}_\mu^{(k-1)} - \lambda\mathcal{A}_\mu^{(k)})\psi_k. \tag{2.25}$$

Here $\lambda$ is related to the cosmological constant $\Lambda$ and $1/g$, the (A)dS curvature scale, as

$$\Lambda = \frac{1}{2}(D-1)(D-2)\lambda, \qquad \lambda = \pm g^2. \tag{2.26}$$

The sign on $\lambda$ depends on if the spacetime is de Sitter or Anti-de Sitter. Accordingly, we will continue writing in terms of the parameter $\lambda$ itself so we capture both cases. Additionally, $\mathcal{A}_\mu^{(k)}$ and $\mathcal{A}^{(k)}$ are functions of the spins $a_\mu$:

$$\mathcal{A}^{(k)} = \sum_{\substack{\nu_1,\cdots,\nu_k=1 \\ \nu_1<\cdots<\nu_k}}^{n-1+\epsilon} a_{\nu_1}^2 \cdots a_{\nu_k}^2 \,, \qquad \mathcal{A}_\mu^{(k)} = \sum_{\substack{\nu_1,\cdots,\nu_k=1 \\ \nu_1<\cdots<\nu_k \\ \nu_i\neq\mu}}^{n-1+\epsilon} a_{\nu_1}^2 \cdots a_{\nu_k}^2 \,. \tag{2.27}$$

The functions $U_\mu$, $A_\mu^{(k)}$, $U$, and $A^{(k)}$ used in (2.24) are given by,

$$A_\mu^{(k)} = \sum_{\substack{\nu_1,\cdots,\nu_k=1 \\ \nu_1<\cdots<\nu_k,\ \nu_i\neq\mu}}^{n} x_{\nu_1}^2 \cdots x_{\nu_k}^2, \quad A^{(k)} = \sum_{\substack{\nu_1,\cdots,\nu_k=1 \\ \nu_1<\cdots<\nu_k}}^{n} x_{\nu_1}^2 \cdots x_{\nu_k}^2,$$

$$U_\mu = \prod_{\substack{\nu=1 \\ \nu\neq\mu}}^{n} (x_\nu^2 - x_\mu^2), \quad U = \prod_{\substack{\mu,\nu=1 \\ \mu<\nu}}^{n} (x_\mu^2 - x_\nu^2) = \det A_\mu^{(j)}. \tag{2.28}$$

The function $X_\mu$ depends only on the single coordinate $x_\mu$, via

$$X_\mu = \frac{-\lambda x_\mu^2 - 1}{(-x_\mu^2)^\epsilon} \prod_{k=1}^{n-1+\epsilon} (a_k^2 - x_\mu^2) + 2M\delta_{\mu,n} \left(-ix_\mu\right)^{1-\epsilon}. \tag{2.29}$$

Because these spacetimes possess a principal tensor [21], they also possess a set of Killing tensors $k_{(j)}^{ab}$, $j = 0,..,n-1$ and Killing vectors $l_{(j)}^a$, $j = 0,...,n+\epsilon-1$. Explicit expressions for these are given by,

$$l_{(j)} = \partial_{\psi_j}, \tag{2.30}$$

$$k_{(j)} = \sum_{\mu=1}^{n} A_\mu^{(j)} \left[ \frac{X_\mu}{U_\mu} \partial_{x_\mu}^2 + \frac{U_\mu}{X_\mu} \left( \sum_{k=0}^{n-1+\epsilon} \frac{(-x_\mu^2)^{n-1-k}}{U_\mu} \partial_{\psi_k} \right)^2 \right] + \epsilon \frac{A_{(j)}}{A_{(n)}} \partial_{\psi_n}^{\,2} \,. \tag{2.31}$$

For $j = 0$, the Killing tensor is just the inverse metric $k_{(0)}^{ab} = g^{ab}$. These Killing objects satisfy the following equations,

$$\nabla^a l_{(j)}^b + \nabla^b l_{(j)}^a = 0, \qquad \nabla^{(a} k_{(j)}^{bc)} = 0\,. \tag{2.32}$$

These objects guarantee exactly enough conserved quantities to allow for the separation of variables in the wave equation. Since $k_{(j)}$ and $l_{(j)}$ are generated by the same principal tensor, the operators $-\nabla_a k_{(j)}^{ab} \nabla_b$ and $-i l_{(j)}^a \nabla_a$ mutually commute. Consequently they have a common eigenfunction $\Phi$. Accordingly, we can write,

$$-\nabla_a k_{(j)}^{ab} \nabla_b \Phi = K_j \Phi, \quad -i l_{(j)}^a \nabla_a \Phi = L_j \Phi\,, \tag{2.33}$$

where the $2n + 2\epsilon$ eigenvalues are written as $K_j$ and $L_j$. Since $k_0$ is the metric, $K_0$ is just the mass of the scalar field; we study the massless Klein-Gordon equation so will often set this parameter to zero. These conserved quantities allow the multiplicative separation ansatz

$$\Phi = \prod_{\mu=1}^{n} R_\mu \prod_{k=0}^{n-1+\epsilon} \exp(iL_k\psi_k), \tag{2.34}$$

where each function $R_\mu$ depends only on one coordinate $x_\mu$, and $R_\mu = R_\mu(x_\mu)$. The authors of [39] showed that equations (2.33) are equivalent to the conditions,

$$X_\mu R_\mu^{''} + \left(X_\mu' + \frac{\epsilon X_\mu}{x_\mu}\right) R_\mu' + \frac{\chi_\mu}{X_\mu} R_\mu = 0, \tag{2.35}$$

where

$$\chi_\mu = X_\mu \sum_{j=0}^{n-1+\epsilon} K_j(-x_\mu^2)^{n-1-j} - \left[\sum_{j=0}^{n-1+\epsilon} L_j(-x_\mu^2)^{n-1-j}\right]^2. \tag{2.36}$$

We find the radial separated wave equation by substituting $x_\mu = x_n = ir$ in (2.35). We find, for the radial wavefunction $R(r)$,

$$- X_r R^{''}(r) - \left(X_r' + \frac{\epsilon X_r}{r}\right) R'(r) + \frac{\chi}{X_r} R_r(r) = 0, \tag{2.37}$$

where $\chi = \chi_n$ and

$$X_r = -\frac{1 - \lambda r^2}{r^{2\epsilon}} \prod_{k=1}^{n+\epsilon-1} (a_k^2 + r^2) + 2Mr^{1-\epsilon} \equiv -\Delta. \tag{2.38}$$

The definition $X_r \equiv -\Delta$ is to remind us that $X_r$ is the generalization of the function $\Delta$, that appears for example in the $4D$ Klein-Gordon equation (2.1), to general dimension and nonzero cosmological constant. We have also chosen the overall factor in (2.37) to match with the $4D$ Klein-Gordon equation (2.1).

## 3 Generalizing the Generators

We now begin to generalize the hidden conformal results of section 2.1 to general dimension black holes with arbitrary spin and arbitrary cosmological constant (A.4). In the process, we will highlight the relationship between the hidden conformal symmetry generators $H_\pm, H_0, \bar{H}_\pm, \bar{H}_0$ and the Killing tower as reviewed in section 2.

### 3.1 Conformal coordinates

We begin by generalizing the conformal coordinates (2.3) and (2.10) so that we can include cosmological constants as in [15] as well as examine higher dimensions [3].

We will want coordinates which include the $r$ and $t$ directions. In analogy with the five-dimensional case (2.10), we will always focus on only one further angle. Eventually we will choose this angle to match one of the Boyer-Lindquist angles $\phi_\mu$ as in (A.4) (since they are periodic up to $2\pi$), but for now we will leave the choice of angle arbitrary, and only insist that it be a Killing direction. We will call this unfixed angle $\psi$. Accordingly, we adopt conformal coordinates $(w^\pm, y)$ defined in terms of $(t, r, \psi)$ by

$$
\begin{aligned}
w^+ &= g(r)h(r)e^{2\pi T_R \psi - 2K_R t}, \\
w^- &= g(r)h(r)e^{2\pi T_L \psi - 2K_L t}, \\
y &= g(r)e^{\pi(T_R + T_L)\psi - (K_R + K_L)t}.
\end{aligned}
\tag{3.1}
$$

Here, $g(r)$ and $h(r)$ are undetermined functions of $r$, while $T_{L,R}$ and $K_{L,R}$ are constants which will be fixed by the black hole geometry (and the choice of $\psi$ angle). For physical clarity in the discussion that follows, however, we will temporarily restrict ourselves to a particular coordinate choice. First, we choose the timelike Boyer-Lindquist coordinate $t$ (A.4), since our system is asymptotically static. Next, just as in the five-dimensional case [10, 16], we also turn on the angular momentum conjugate to only one single $\phi_\mu$. We will set this particular index $\mu = \star$ in what follows. We pick among the Boyer-Lindquist angles $\phi_\mu$ because they are all identified up to $2\pi$, as previously discussed in 3.1. All other momenta conjugate to Killing vector directions are turned off.

The two coordinate systems $(t, r, \phi_\star)$ and $(w^\pm, y)$ have different utilities. As we now explain, the Boyer-Lindquist coordinates $(t, r, \phi_\star)$ show the thermal nature of the black hole background, while the conformal coordinates $(w^\pm, y)$ exhibit any conformal structure of the black hole horizon. We know from [41] that three-dimensional black holes can be constructed from portions of $AdS_3$ (such as the upper half-plane) via a coordinate identification that imbues the spacetime with a quotient structure. The importance of the Boyer-Lindquist-like coordinates $(t, r, \phi_\star)$ is that this periodic coordinate identification $\phi_\star \sim \phi_\star + 2\pi$ is evident. In addition, these are the coordinates that are generally used to define energy and momentum (that is, for eigenmode $\Phi = R(r)S(\theta)e^{i(k\phi_\star - \omega t)}$ the eigenvalues of $i\partial_t$ and $-i\partial_{\phi_\star}$, respectively).

On the other hand, as mentioned in Section 2.1.1, near the bifurcation surface $w^\pm = 0$, the conformal coordinates $(w^\pm, y)$ transform a constant $\theta$ slice of the Kerr metric (A.1) into warped $AdS_3$ (2.5). Furthermore, the form of the coordinates (3.1) are adapted from the well-understood case of the BTZ black hole. As discussed in [37], the boundary of $AdS_3$ inherits a natural set of null coordinates (analogous to $w^\pm$) in which the boundary CFT is in its vacuum state. Thus the reason for the exponential structure of the coordinate transformation (3.1) between "vacuum" coordinates and "thermal" coordinates is clear: it is just the coordinate transformation between the Minkowski vacuum and the Rindler wedge. This point is discussed for example in [8, 16, 37].

There is an ambiguity in defining the exponential factors of the conformal coordinates in

(3.1). A priori, we could consider any combination of the coordinates $(t, \phi_\star)$

$$
\begin{aligned}
w^+ &= g(r)h(r)e^{a\phi_\star + bt}, \\
w^- &= g(r)h(r)e^{c\phi_\star + dt}, \\
y &= g(r)e^{((a+c)\phi_\star + (b+d)t)/2},
\end{aligned}
\tag{3.2}
$$

leaving us with the four parameter family $(a, b, c, d)$. First, choosing that the metric (A.4) has the expansion (2.5) near the bifurcation surface $w^\pm = 0$ fixes two of the parameters $(a, b, c, d)$.

As discussed in [15], the other two parameters are fixed by insisting that the linear combination of $(t, \phi_\star)$ (or rather $(\partial_t, \partial_{\phi_\star})$) that are of physical interest are

$$
\zeta^\pm = \kappa_\pm \left( \partial_t + \sum_{\mu=1}^{n+\epsilon-1} \Omega_{\mu,\pm} \partial_{\phi_\mu} \right).
\tag{3.3}
$$

Here $\kappa_\pm$ is the surface gravity at the inner and outer horizons, and $\Omega_{\mu,\pm}$ with the angular velocity of the inner and outer horizon with respect to angle $\phi_\mu$. Fixing an angle $\phi_\mu = \phi_\star$ means that all terms $\partial_{\phi_\mu}\Phi$ will vanish for $\mu \neq \star$. The parameter $\epsilon$ distinguishes between even and odd dimensions: $\epsilon = 0$ for $D$ even and $\epsilon = 1$ for $D$ odd, and $n$ is defined such that $D = 2n + \epsilon$. The generators (3.3) are interesting from a thermodynamic point of view. Wald noted [42] that the entropy of a general bifurcate Killing horizon is equal to the integrated Noether charge associated to the Killing field vanishing on that surface. The generators (3.3) are precisely those vanishing on the inner and outer horizons $r_\pm$. We would like to build the set of 6 locally defined $SL(2, R) \times SL(2, R)$ symmetry generators $(H, \bar{H})$ defined in (2.6) from conformal coordinates (3.1). If we do this correctly, the quadratic casimir (2.8) should match the radial Klein-Gordon operator (Laplacian) in (2.37). We will now see that demanding the Casimir $\mathcal{H}^2$ is proportional to (2.37) will constrain the conformal coordinates further.

We find the conformal coordinates (3.1) obey the relations

$$
\begin{aligned}
\frac{w^+ w^-}{y^2} &= h^2, \\
\frac{w^+}{w^-} &= e^{-2\pi(T_-\psi - K_- t)}, \\
w^+ w^- + y^2 &= g^2(1 + h^2)e^{2\pi(T_+\psi - K_+ t)},
\end{aligned}
\tag{3.4}
$$

where we have defined $T_\pm = T_L \pm T_R$ and $\pi K_\pm = K_L \pm K_R$. Taking partial derivatives with respect to $w^+$, $w^-$ and $y$ on each of these relations, and doing some algebra, we find

$$
\begin{aligned}
y\partial_y &= \frac{2T_-}{\Omega}\left(1 + \frac{hg'}{h'g}\right)\partial_t + \frac{2K_-}{\Omega}\left(1 + \frac{hg'}{h'g}\right)\partial_\psi - \frac{h}{h'}\partial_r, \\
w^+\partial_+ &= \frac{1}{\Omega}\left(T_+ - T_-\frac{hg'}{h'g}\right)\partial_t + \frac{1}{\Omega}\left(K_+ - K_-\frac{hg'}{h'g}\right)\partial_\psi + \frac{h}{2h'}\partial_r, \\
w^-\partial_- &= \frac{1}{\Omega}\left(-T_+ - T_-\frac{hg'}{h'g}\right)\partial_t + \frac{1}{\Omega}\left(-K_+ - K_-\frac{hg'}{h'g}\right)\partial_\psi + \frac{h}{2h'}\partial_r,
\end{aligned}
\tag{3.5}
$$

where we have now defined

$$\Omega \equiv 2\pi(T_+K_- - T_-K_+) = 4\left(T_RK_L - T_LK_R\right). \tag{3.6}$$

Since, by inspection, we can see that the radial Klein-Gordon operator in (2.37) contains no cross terms of the form $\partial_t\partial_r$ or $\partial_\psi\partial_r$, we can further constrain our conformal coordinates. By plugging the partial derivative expressions (3.5) into the quadratic Casimir (2.8) as found from the $H$ generators, we find the term

$$\frac{-T_-}{\Omega hh'}\left[h^2 + \left(h^2+1\right)\frac{hg'}{h'g}\right]\partial_t\partial_r, \tag{3.7}$$

and a similar term for $\partial_\psi\partial_r$. Since both of these coefficients must vanish, in order for the quadratic Casimir to match the radial Klein-Gordon operator, we must have

$$\frac{hh'}{h^2+1} = \frac{-g'}{g}. \tag{3.8}$$

This equation is satisfied if we set

$$g^2(h^2+1) = C \tag{3.9}$$

for any constant $C$. In the four- and five-dimensional cases, $C$ was set to 1; since it is an overall scale in the conformal coordinates (3.2), we will keep this fixing below.

Under the restriction (3.9), the Casimir built from the $H$ generators becomes

$$\begin{aligned}
\mathcal{H}^2 =& \frac{h^2+1}{4(h')^2}\partial_r^2 + \left(\frac{1+h^2}{4hh'}\left[\frac{(h')^2 - h''h}{(h')^2}\right] + \frac{h}{2h'}\right)\partial_r + \left(\frac{T_-^2}{\Omega^2(h^2+1)} - \frac{T_+^2}{\Omega^2h^2}\right)\partial_t^2 \\
&+ \left(\frac{2T_-K_-}{\Omega^2(h^2+1)} - \frac{2T_+K_+}{\Omega^2h^2}\right)\partial_t\partial_\psi + \left(\frac{K_-^2}{\Omega^2(h^2+1)} - \frac{K_+^2}{\Omega^2h^2}\right)\partial_\psi^2.
\end{aligned} \tag{3.10}$$

## 3.2 Matching the $r$-derivative Pieces of the Separated Klein-Gordon Equation

We aim to match the quadratic Casimir (3.10) to the radial Klein-Gordon operator as in (2.37)-(2.38). We begin by comparing the $\partial_r^2$ and $\partial_r$ terms, deferring matching of the non-derivative terms to section 3.3.

Matching the $r$-derivative terms fixes the radial dependence $h(r)$ of the conformal coordinates (3.1) for the general dimension black hole. In the case of asymptotically flat black holes in 4 and 5 dimensions, we can directly match the coefficients of $\partial_r^2$ and $\partial_r$ from the Klein-Gordon equation as in (2.37). Matching the double and single $r$-derivative pieces, we find

$$\Delta = \frac{h^2+1}{4h'^2}, \qquad \Delta' + \frac{\epsilon\Delta}{r} = \frac{1+h^2}{4hh'}\frac{d}{dr}\left(\frac{h}{h'}\right) + \frac{h}{2h'}. \tag{3.11}$$

Solving these two equations simultaneously recovers an expression for $h(r)$, but also requires that the leading order piece of $\Delta(r) \sim r^2$. Thus, matching directly the coefficients for 3.11 is only possible for asymptotically flat black holes in 4 and 5 dimensions.[6]

---

[6]We can see that allowing for a nonzero cosmological constant in the metric factor $\Delta = -X_r$ as in (2.38) increases the leading power of $r$, so asymptotically flat black holes in 4 or 5 dimensions are the only cases where we can match directly the coefficients of (3.11).

Consequently, we will multiply the Klein-Gordon equation (2.37) by an overall scalar factor $s$. Accordingly, we must solve instead the equations

$$s\Delta = \frac{h^2+1}{4h'^2}, \qquad s\left(\Delta' + \frac{\epsilon\Delta}{r}\right) = \frac{1+h^2}{4hh'}\frac{d}{dr}\left(\frac{h}{h'}\right) + \frac{h}{2h'}. \qquad (3.12)$$

We begin by equating the ratios

$$\frac{\Delta' + \frac{\epsilon\Delta}{r}}{\Delta} = \frac{\frac{1+h^2}{4hh'}\frac{d}{dr}\left(\frac{h}{h'}\right) + \frac{h}{2h'}}{\frac{h^2+1}{4h'^2}}. \qquad (3.13)$$

Solving (3.13) gives

$$\Delta = \frac{r^{-\epsilon}c_1 h(1+h^2)}{h'}, \qquad (3.14)$$

which we can use to solve for the radial function $h(r)$. We find:

$$h^2 = \frac{e^I}{1-e^I}, \qquad (3.15)$$

where

$$I = 2c_1 \int \frac{r^{-\epsilon}}{\Delta(r)}dr. \qquad (3.16)$$

Using (3.14), we can easily find the scalar function $s$ in terms of $h$:

$$s \equiv \frac{h^2+1}{4h'^2\Delta} = \frac{r^\epsilon}{4c_1 h'h} = \frac{\Delta r^{2\epsilon}}{4c_1^2 h^2(h^2+1)}. \qquad (3.17)$$

In principle, we have now fixed the radial dependence of the conformal coordinates. However, the form of the radial function $\Delta = -X_r$ (2.38) is sufficiently simple that we can actually compute the integral (3.16) for the general spin (A)dS black hole in arbitrary dimensions. Since $\Delta = 0$ defines the horizon locations (only two of which are real and positive for flat space or AdS; de Sitter can also have cosmological horizons), we will now rewrite it in terms of its roots. Again using $\epsilon = 0$ for even dimensions, $\epsilon = 1$ for odd dimensions, and setting $\sigma = 0$ for flat space and $\sigma = 1$ for (A)dS, we can write

$$\Delta = r^{-2\epsilon}\prod_{i=1}^{2N_{\epsilon,\sigma}}(r-r_i), \qquad N_{\epsilon,\sigma} = n-1+\epsilon+\sigma. \qquad (3.18)$$

where $2N_{\epsilon,\sigma} = 2(n-1+\epsilon+\sigma)$ is the number of roots. We will also make use of the handy relation

$$\Delta'(r_i) = r_i^{-2\epsilon}\prod_{j=1,j\neq i}^{2N_{\epsilon,\sigma}}(r_i-r_j). \qquad (3.19)$$

Let's look at the even case first, when $\epsilon = 0$. We find

$$I = 2c_1 \int \frac{1}{\prod_{i=1}^{2N_{0,\sigma}}(r-r_i)} dr = 2c_1 \left( \sum_{i=1}^{2N_{0,\sigma}} \frac{\log(r-r_i)}{\prod_{j=1,j\neq i}^{2N_{0,\sigma}}(r_i-r_j)} + c_2 \right)$$
$$= 2c_1 \left( \sum_{i=1}^{2N_{0,\sigma}} \frac{\log(r-r_i)}{\Delta'(r_i)} + c_2 \right). \tag{3.20}$$

For the odd case ($\epsilon = 1$) we have

$$I = 2c_1 \int \frac{r}{\prod_{i=1}^{2N_{1,\sigma}}(r-r_i)} dr = 2c_1 \left( \sum_{i=1}^{2N_{1,\sigma}} \frac{r_i \log(r-r_i)}{\prod_{j=1,j\neq i}^{2N_{1,\sigma}}(r_i-r_j)} + c_2 \right)$$
$$= 2c_1 \left( \sum_{i=1}^{2N_{1,\sigma}} \frac{\log(r-r_i)}{r_i \Delta'(r_i)} + c_2 \right). \tag{3.21}$$

Combining (3.20) and (3.21), we have

$$I = 2c_1 \left( \sum_{i=1}^{2N_{\epsilon,\sigma}} \frac{\log(r-r_i)}{r_i^\epsilon \Delta'(r_i)} + c_2 \right). \tag{3.22}$$

For the $4D$ and $5D$ cases, it will turn out that we can set $c_2 = 0$. For the large $D$ analysis, we will see that we require a different choice for $c_2$. But for now, setting $c_2 = 0$, we arrive at

$$e^I = \prod_{i=1}^{2N_{\epsilon,\sigma}} (r-r_i)^{\frac{2c_1}{r_i^\epsilon \Delta'(r_i)}}. \tag{3.23}$$

For a $4D$ black hole in flat space, we have only two real roots: the inner and outer horizons $r_\pm$. For this case, $e^I$ simplifies considerably. Since for any polynomial (with isolated roots), $\sum_i \frac{1}{\Delta'(r_i)} = 0$, here we have $\Delta'(r_+) = -\Delta'(r_-)$. Thus, we find

$$e_{4D}^I = \left( \frac{r-r_+}{r-r_-} \right)^{2c_1/\Delta'(r_+)}. \tag{3.24}$$

The exponent here is now just a constant. Since $c_1$ is not yet fixed, we use it to set the exponent to one; that is, we choose

$$c_1 = \frac{1}{2}\Delta'(r_+). \tag{3.25}$$

Using the relation (3.15) we find

$$h_{4D}^2 = \frac{r-r_+}{r_+ - r_-}, \tag{3.26}$$

which indeed matches $w_+ w_- / y^2 = h^2$ for the $4D$ conformal coordinates (2.3). We can additionally check that $s = 1$ for this case.[7]

For a black hole in $6D$ flat space, or for the $4D$ case with a nonzero cosmological constant, there are four roots. We still have the relation $\sum_i \frac{1}{\Delta'(r_i)} = 0$, but our single constant $c_1$ is not enough to set all of the three remaining exponents to integers. In general no further relation exists among the roots, and so $e^I$ and $h^2$ have branch cuts. These branch cuts prevent matching the remaining terms in the Casimir (3.10) to the Klein-Gordon equation. However, we can again choose the constant $c_1 = \frac{1}{2} \Delta'(r_+)$ which sets the exponent for the outer horizon, $r_+$, to one. If we then expand the rest of the terms near the outer horizon, we can recover $e^I$ as a ratio of polynomials as needed. As we will see below, even this near-horizon limit will not allow us to completely match the quadratic Casimir; an $\omega^2$ error term is still present. However, since this error term does not become large as $r \to r_\pm$, while the other terms in the wave equation have singularities at the horizons. Accordingly, a near-horizon limit will make the $\omega^2$ term small in comparison. Similar behavior occurs for all $D \geq 6$.

### 3.3 Matching the Killing Vector Directions

We now build the matching equations for the coefficients of the terms without $r$-derivatives, choosing our coordinates in the manner discussed in Section 3.1. That is, we choose the timelike Boyer-Lindquist coordinate $t$ and a single Boyer-Lindquist angular direction $\phi_\mu$ (A.4). To highlight that we are choosing one particular angle $\phi_\mu$ we often call the index $\mu = \star$.

The remaining piece we wish to match in the Klein-Gordon equation (2.37) is

$$
- s \frac{\chi}{\Delta} = s \sum_{j=0}^{n-1+\epsilon} K_j (r^2)^{n-1-j} + \frac{s}{\Delta} \left[ \sum_{j=0}^{n-1+\epsilon} L_j (r^2)^{n-1-j} \right]^2 , \tag{3.27}
$$

Here $\chi$ is obtained by setting $\mu = n = r$ in (2.36). To proceed, we need to identify the $L_j$, so we write our Klein-Gordon scalar in Fourier modes as

$$
\Phi \propto e^{im\phi_\star - i\omega t} \propto e^{im \sum \frac{\partial \phi_\mu}{\partial \psi_k} \psi_k - i\omega \frac{\partial t}{\partial \psi_k} \psi_k} . \tag{3.28}
$$

Using the coordinate relations (2.25) and the definition

$$
L_k \Phi = -i \frac{\partial \Phi}{\partial \psi_k} , \tag{3.29}
$$

we find

$$
L_k = m a_\star \left( \lambda \mathcal{A}_\star^{(k)} - \mathcal{A}_\star^{(k-1)} \right) - \omega \mathcal{A}^{(k)} . \tag{3.30}
$$

Here, the $\mathcal{A}$ and $\mathcal{A}_\star$ are defined as in (2.27), except we have additionally used

$$
\mathcal{A}_\star^{(0)} = \mathcal{A}^{(0)} = 1, \qquad \mathcal{A}_\star^{-1} = 0. \tag{3.31}
$$

---

[7]For a black hole in $5D$ flat space a similar trick works; although there are 4 real roots, they are just $\pm r_\pm$, so again we are able to set the exponent to an integer by choosing $c_1 = r_+^2 - r_-^2$; again we find the $5D$ conformal coordinates (2.10) match with this choice. In this case we find $s = 1/4$.

In addition to allowing only the $\omega$ and $m$ momenta, we have one further adjustment to make to our radially separated wave equation (2.37). We have already multiplied by $s$ (3.17), but now we will want to isolate the separation constants that come from the Killing tensors. Much as in [8, 10], we will not ask the Casimir to reproduce these terms, moving them instead to the right-hand side of our Klein-Gordon equation.

As we will see in detail in sections 3.5 and 3.6, for both the $4D$ and $5D$ cases, the Killing tensor separation constant terms are shifted from the $K_1$ values given by (2.33).[8] Explicitly, we allow the shift

$$K'_k = K_k - \sum_{i=0}^{n+\epsilon-1} \sum_{i=0}^{n+\epsilon-1} Q_k^{ij} L_i L_j \,. \tag{3.32}$$

There are two special cases which we will not shift: first, $K_n = L_n^2/\mathcal{A}^{(n)}$, present only in odd dimensions, is actually already built just from the $L_n$. Next, $K_0$, since it is built from $k_{(0)}^{ab} = g^{ab}$, is just the Klein-Gordon mass$^2$. As we are studying the massless equation, we will not want to shift this constant.

Expanding $\chi$ explicitly as in (3.27), and moving the $K'_n$ terms to the right hand side, we rewrite the Klein-Gordon equation (2.37) as

$$\Delta R'' + \left( \Delta' + \frac{\epsilon\Delta}{r} \right) R' + \sum_{i=0}^{n-1+\epsilon} \sum_{j=0}^{n-1+\epsilon} L_i L_j \left( \frac{1}{\Delta} r^{2(2n-2-i-j)} + \sum_{k=1}^{n-1+\epsilon} Q_k^{ij} r^{2(n-1-k)} \right) = -R \sum_{k=0}^{n-1} K'_k r^{2(n-1-k)} \,. \tag{3.33}$$

Here we have defined $Q_n^{ij} \equiv \epsilon \delta_n^i \delta_n^j / \mathcal{A}^{(n)}$ for compactness.

Since we have already matched the $r$-derivative pieces, our goal is to match the last term on left hand side of (3.33), multiplied by $s$ as in (3.17), with the $\partial_t$ and $\partial_\psi = \partial_{\phi_\star}$ terms in (3.10). Multiplying both of these expressions by $h^2(h^2+1)$, we obtain

$$\frac{1}{4c_1^2} \sum_{i=0}^{n-1+\epsilon} \sum_{j=0}^{n-1+\epsilon} L_i L_j \left( r^{2(2n-2-i-j+\epsilon)} + \Delta \sum_{k=1}^{n-1+\epsilon} Q_k^{ij} r^{2(n-1-k+\epsilon)} \right)$$

$$= \frac{1}{\Omega^2} \left[ - \left( T_-^2 h^2 - T_+^2 (h^2+1) \right) \omega^2 + \left( T_- K_- h^2 - T_+ K_+ (h^2+1) \right) 2m\omega - \left( K_-^2 h^2 - K_+^2 (h^2+1) \right) m^2 \right] \,. \tag{3.34}$$

The $L_k$ here are given by (3.30), and consequently depend only on $m$ and $\omega$. We will need to match the $m^2$ terms and the $m\omega$ terms, but the $\omega^2$ terms may be partially absorbed by the generalization of the near-region limit from section 2.1.1.

---

[8]The $4D$ and $5D$ cases also have another simplification not present in the general case: since only $K_1$ is nonzero in those cases, and since $s$ itself is also a constant, the term on the right-hand side is also a constant (no powers of $r$ remain).

The $m^2$ terms require

$$
-\frac{1}{\Omega^2}\left(K_-^2 h^2 - K_+^2(h^2+1)\right)
$$

$$
= \frac{a_\star^2}{4c_1^2}\sum_{i=0}^{n-1+\epsilon}\sum_{j=0}^{n-1+\epsilon}\left(\lambda\mathcal{A}_\star^{(i)} - \mathcal{A}_\star^{(i-1)}\right)\left(\lambda\mathcal{A}_\star^{(j)} - \mathcal{A}_\star^{(j-1)}\right) r^{2(n-1+\epsilon)}\left(r^{2(n-1-i-j)} + \Delta\sum_{k=1}^{n-1+\epsilon} Q_k^{ij} r^{-2k}\right),
$$

(3.35)

while the $m\omega$ terms need

$$
\frac{2}{\Omega^2}\left(T_- K_- h^2 - T_+ K_+ (h^2+1)\right) =
$$

$$
\frac{-a_\star}{4c_1^2}\sum_{i=0}^{n-1+\epsilon}\sum_{j=0}^{n-1+\epsilon}\left(\left[\lambda\mathcal{A}_\star^{(i)} - \mathcal{A}_\star^{(i-1)}\right]\mathcal{A}^{(j)} + \left[\lambda\mathcal{A}_\star^{(j)} - \mathcal{A}_\star^{(j-1)}\right]\mathcal{A}^{(i)}\right)
$$

(3.36)

$$
\times r^{2(n-1+\epsilon)}\left(r^{2(n-1-i-j)} + \Delta\sum_{k=1}^{n-1+\epsilon} Q_k^{ij} r^{-2k}\right).
$$

From these two equations, we can more clearly see the trouble with $h^2$ being non-polynomial. If $e^I$ (3.23), and thus $h^2$, is itself a ratio of polynomials with integer powers of $r$, then it is possible to choose $T_\pm$, $K_\pm$, and $Q_k^{ij}$, all functions of only the black hole parameters, so that the matching equations (3.35) and (3.36) are both satisfied. Then, we can take an appropriate near-region limit, in both $\omega$ and $r$, to eliminate the remaining $\omega^2$ terms.

However, we really only need a near-region limit for $4D$ and $5D$ black holes with flat asymptotics. If we add a cosmological constant, or if we go up in dimension, then $e^I$ contains non-integer powers, and we must first take a near-horizon limit in $r$ alone before taking the near-region limit. Since the near-horizon limit will cause the singular terms which show up in the Casimir to be larger than the nonsingular $\omega^2$ term, the near-region limit is redundant there.

## 3.4 Sandwiching the $H$s: Building towards a Tensor Equation

Now that we have made the form of our conformal coordinates (3.1) more explicit, we will work towards a tensor equation which will exhibit the relationship between the quadratic Casimir $\mathcal{H}^2$ and the separated radial equation (2.37). Beginning with the radial equation (3.33), we have already matched $\mathcal{H}^2/s$ to the left hand side of this equation, in sections 3.2 and 3.3. Up to the near-horizon limit, we have

$$
\mathcal{H}^2\Phi = -s\sum_{k=0}^{n-1}\left(K_k - \sum_{i=0}^{n+\epsilon-1}\sum_{i=0}^{n+\epsilon-1} Q_k^{ij} L_i L_j\right) r^{2(n-1-k)}\Phi.
$$

(3.37)

We can rewrite this equation as a tensor operator acting on $\Phi$:

$$
\left(-H_0^a\nabla_a H_0^b\nabla_b + \frac{1}{2}H_1^a\nabla_a H_{-1}^b\nabla_b + \frac{1}{2}H_{-1}^a\nabla_a H_1^b\nabla_b\right)\Phi
$$

$$
= -s\sum_{k=0}^{n-1} r^{2(n-1-k)}\left(-\nabla_a k_{(k)}^{ab}\nabla_b + \sum_{i=0}^{n+\epsilon-1}\sum_{i=0}^{n+\epsilon-1} Q_k^{ij} l_{(i)}^a\nabla_a l_{(j)}^b\nabla_b\right)\Phi
$$

(3.38)

Here we have used the eigenequation (2.33).

We want to propose a tensor equation $T^{ab} = 0$, which enforces the result $\nabla_a T^{ab} \nabla_b \Phi$. First, though, we need to rearrange the pieces of the form $H_i^a \nabla_a H_j^b \nabla_b$ to instead be in the sandwiched form $\nabla_a \left( H_i^a H_j^b \nabla_b \right)$. This rearrangement produces an extra term, since

$$H_i^a \nabla_a H_j^b \nabla_b = \nabla_a \left( H_i^a H_j^b \nabla_b \right) - \left( \nabla_a H_i^a \right) H_j^b \nabla_b. \tag{3.39}$$

We also need to reorder $l_{(i)}^a \nabla_a l_{(j)}^b \nabla_b = \nabla_a l_{(i)}^a l_{(j)}^b \nabla_b$, which is satisfied as the $l_{(i)}$ are all Killing vectors. Last, we need to pull the factor of $sr^{2(n-1-k)}$ inside, which produces an extra term:

$$\sum_{k=0}^{n-1} (\nabla_a sr^{2(n-1-k)}) \left( -k_{(k)}^{ab} \nabla_b + \sum_{i=0}^{n+\epsilon-1} \sum_{i=0}^{n+\epsilon-1} Q_k^{ij} l_{(i)}^a l_{(j)}^b \nabla_b \right) \Phi. \tag{3.40}$$

Since $s$ only depends on $r$, $k_{(k)}^{ab}$ is symmetric, and $r$ is not a killing vector direction, this term reduces to

$$-\sum_{k=0}^{n-1} \left( \partial_r \left[ sr^{2(n-1-k)} \right] \right) k_{(k)}^{rb} \nabla_b \Phi. \tag{3.41}$$

We thus propose the tensor equation

$$-H_0^a H_0^b + \frac{1}{2} H_1^a H_{-1}^b + \frac{1}{2} H_{-1}^a H_1^b = -s \sum_{k=0}^{n-1} r^{2(n-1-k)} \left( -k_{(k)}^{ab} + \sum_{i=0}^{n+\epsilon-1} \sum_{i=0}^{n+\epsilon-1} Q_k^{ij} l_{(i)}^a l_{(j)}^b \right) + E^{ab}, \tag{3.42}$$

where the error term $E^{ab}$ satisfies

$$\nabla_a E^{ab} \nabla_b \Phi = - \left( \nabla_a H_0^a \right) H_0^b \nabla_b \Phi + \frac{1}{2} \left( \nabla_a H_1^a \right) H_{-1}^b \nabla_b \Phi + \frac{1}{2} \left( \nabla_a H_{-1}^a \right) H_1^b \nabla_b \Phi$$
$$- \sum_{k=0}^{n-1} \left( \partial_r \left[ sr^{2(n-1-k)} \right] \right) k_{(k)}^{rr} \nabla_r \Phi - \tilde{Q}^{00} \omega^2 \Phi + \tilde{E} \Phi + E_\star \Phi. \tag{3.43}$$

Here, $\tilde{E}$ expresses any errors from the near-horizon limit necessary to make $e^I$ a ratio of polynomial terms. In $4D$ and $5D$ with flat asymptotics, this term should be absent, since no near-horizon limit is needed.

Since we only matched the $m^2$ and $m\omega$ terms in the Klein-Gordon equation to the Casimir, we expect the $\omega^2$ term does not yet match; accordingly, we have included an adjustment $\tilde{Q}^{00}$ (which is allowed to depend on $r$). Additionally, in $D \geq 5$, the choice to concentrate on only one Boyer-Lindquist angle $\phi_\mu = \phi_\star$, means that we should expect error terms along $\phi_\mu$ for all $\mu \neq \star$; we have denoted these contributions by $E_\star$. In particular the condition that $\partial_{\mu \neq = \star} \Phi = 0$ will set $E_\star \Phi = 0$.

We expect the entire error term in (3.43) to vanish, regardless of dimension, when we additionally take a near-region limit; that is, $\nabla_a E^{ab} \nabla_b \Phi$ should be $\mathcal{O}(\omega^2)$. Even more specifically, they will be $\mathcal{O}(\omega^2 r^2)$ or $\mathcal{O}(\omega^2 M^2)$. In order to recover a valid near-region limit of this

form, we may need to set the constants $Q_k^{00}$ appropriately. Since we did not match the $\omega^2$ terms in the Klein-Gordon equation to the Casimir, the constants are as yet unfixed.

As we will show in the next section for $4D$ and $5D$ black holes with flat asymptotics, $s$ is a constant and the error term $\tilde{E}$ indeed vanishes, and $\nabla_a E^{ab} \nabla_b \Phi$ consists of only terms of order $\omega^2$, which describe exactly the near-region limit. The general case is considerably more challenging so we defer it to future work, instead exploring the large $D$ limit in (5).

## 3.5 The Tensor Equation for Kerr in $D = 4$

Here we will use the results from the previous four sections to rederive the conformal coordinates for $4D$ Kerr. In the process, we will find the tensor equation for Kerr as expected from section 3.4.

We begin with $h_{4D}^2$ from (3.26), $s = 1$, and $2c_1 = r_+ - r_-$ from (3.25), all of which arose from matching the Casimir to the $r$-derivative pieces in section 3.2. Our goal is to find the remaining four parameters which specify the conformal coordinates in (3.1): $T_{L,R}$ and $K_{L,R}$. We will use the definitions $T_\pm, K_\pm$ as well as $\Omega$ as in (3.6) from 3.1.

As discussed in section 3.3, to fix these remaining parameters we need to match the $m^2$ terms following (3.35) and the $m\omega$ terms as in (3.36). Using the definitions from (2.27) and (3.31), we find that the only nonzero $\mathcal{A}$ for the $4D$ flat Kerr case are

$$\mathcal{A}^{(0)} = 1, \qquad \mathcal{A}^{(1)} = a^2, \qquad \mathcal{A}_1^{(0)} = 1. \tag{3.44}$$

Since $\lambda = 0$ for flat asymptotics, we can reduce the $m^2$ equation (3.35) to

$$\frac{K_+^2 - K_-^2}{\Omega^2} r - \frac{K_+^2 r_- - K_-^2 r_+}{\Omega^2} = \frac{a^2}{r_+ - r_-} \left(1 + \Delta Q_1^{11}\right). \tag{3.45}$$

Since $\Delta = r^2 - 2Mr + a^2$, we can immediately see that $Q_1^{11} = 0$ in order to match coefficients of $r$ on both sides. Matching the other powers of $r$ requires

$$K_+^2 = K_-^2 = \frac{\Omega^2 a^2}{(r_+ - r_-)^2}. \tag{3.46}$$

Similarly for the $m\omega$ equation (3.36), we find

$$\frac{2}{\Omega^2} \left[T_- K_-(r - r_+) - T_+ K_+(r - r_-)\right] = \frac{2a}{r_+ - r_-} \left[r^2 + \Delta Q_1^{10} + a^2\right], \tag{3.47}$$

where we have used that $Q_k^{ij} = Q_k^{ji}$ is symmetric. Now, matching the $r^2$ coefficients requires $Q_1^{10} = -1$. Matching the remaining terms and satisfying the constraint (3.46) fixes

$$T_+ = \frac{r_+}{2\pi a}, \qquad T_- = \frac{r_-}{2\pi a}, \qquad K_+ = K_- = \frac{1}{2\pi(r_+ + r_-)}, \tag{3.48}$$

up to some overall sign choices. The choices listed here recover

$$T_R = \frac{r_+ - r_-}{4\pi a}, \qquad T_L = \frac{r_+ + r_-}{4\pi a}, \qquad K_R = 0, \qquad K_L = \frac{1}{4M}, \tag{3.49}$$

which match the conformal coordinates in [8].

In addition to the $T_{L,R}$ and $K_{L,R}$, we also found

$$Q_1^{11} = 0, \qquad Q_1^{10} = Q_1^{01} = -1. \tag{3.50}$$

We can thus propose the tensor equation (3.42) for the specific case of $4D$. Recalling that $Q_0^{ij} = 0$ and $k_{(0)}^{ab} = g^{ab}$, we expect

$$- H_0^a H_0^b + \frac{1}{2} H_1^a H_{-1}^b + \frac{1}{2} H_{-1}^a H_1^b = r^2 g^{ab} + k_{(1)}^{ab} + l_{(1)}^a l_{(0)}^b + l_{(0)}^a l_{(1)}^b - Q_1^{00} l_{(0)}^a l_{(0)}^b + E^{ab}. \tag{3.51}$$

In order to find the error term $E^{ab}$, we first compute the cost to sandwich the $H$ generators:

$$-(\nabla_a H_0^a) H_0^b \nabla_b \Phi + \frac{1}{2} (\nabla_a H_1^a) H_{-1}^b \nabla_b \Phi + \frac{1}{2} (\nabla_a H_{-1}^a) H_1^b \nabla_b \Phi = 2r \frac{r^2 - 2Mr + a^2}{r^2 + a^2 \cos^2 \theta} \partial_r \Phi. \tag{3.52}$$

Next we compute the cost to move $sr^{2(n-1-k)}$:

$$-\sum_{k=0}^{1} \left( \partial_r \left[ sr^{2(1-k)} \right] \right) k_{(k)}^{rr} \nabla_r \Phi = -2rg^{rr} \partial_r \Phi = -2r \frac{r^2 - 2Mr + a^2}{r^2 + a^2 \cos^2 \theta} \partial_r \Phi. \tag{3.53}$$

We can rewrite these residuals as $\pm \nabla_a \left( r^2 g^{ab} \nabla_b \Phi \right)$, perhaps unsurprisingly as this term matches the the contribution from single $r$-derivative terms in the Casimir; we chose the factor $s$ to ensure those terms would be correctly reproduced when the double $r$-derivative coefficient was fixed to $\Delta$.

Consequently, these two costs cancel, leaving us with an error term in (3.43) of the form

$$\nabla_a E^{ab} \nabla_b \Phi = -\tilde{Q}^{00} \omega^2 \Phi. \tag{3.54}$$

If we additionally choose to adjust our separation constant using $Q_1^{00} = a^2$ in (3.51), then we find the error term becomes

$$\tilde{Q}^{00} = 4M^2 + 2Mr + r^2, \qquad E^{ab} = \delta_t^a \delta_t^b \tilde{Q}^{00}. \tag{3.55}$$

Indeed, this error term vanishes in a near-region limit, and the explicit form of (3.51) can be verified as a tensor equation. Additionally, we note that the same exact equation holds for the $\bar{H}$ generators, as expected since their Casimirs are designed to match.

## 3.6 The Tensor Equation for Myers-Perry in D=5

We will now rederive the conformal coordinates for $5D$ Myers-Perry, and again find a tensor equation as in section 3.4 along the way.

We begin by using (3.23) with a choice of $c_1 = r_+^2 - r_-^2$ to find

$$e^{I_{5D}} = \frac{r^2 - r_+^2}{r^2 - r_-^2}, \qquad h_{5D}^2 = \frac{r^2 - r_+^2}{r_+^2 - r_-^2}. \tag{3.56}$$

and, from (3.17), $s = 1/4$. The nonzero $\mathcal{A}$ from (2.27) and (3.31) become

$$\mathcal{A}^{(0)} = 1\,, \qquad \mathcal{A}^{(1)} = a_1^2 + a_2^2\,, \qquad \mathcal{A}^{(2)} = a_1^2 a_2^2\,, \tag{3.57}$$

while the $\mathcal{A}_{\mu=\star}$ become

$$\mathcal{A}_\star^{(0)} = 1\,, \qquad \mathcal{A}_\star^{(1)} = \frac{a_1^2 a_2^2}{a_\star^2}\,. \tag{3.58}$$

We will also want to use the following relationships valid for $5D$ asymptotically flat black holes:

$$r_+^2 + r_-^2 = 2M - a_1^2 - a_2^2\,, \qquad r_+^2 r_-^2 = a_1^2 a_2^2\,. \tag{3.59}$$

Again using $\lambda = 0$ for flat asympotics, the $m^2$ equation (3.35) becomes

$$-\frac{1}{\Omega^2}\left(K_-^2(r^2 - r_+^2) - K_+^2(r^2 - r_-^2)\right)$$
$$= \frac{a_\star^2}{4(r_+^2 - r_-^2)}\left[(r^2 + \Delta r^2 Q_1^{11}) + \frac{a_1^2 a_2^2}{a_\star^2}\left(2 + 2\Delta r^2 Q_1^{12}\right) + \frac{a_1^4 a_2^4}{a_\star^4}\left(r^{-2} + \Delta r^2 Q_1^{22} - \frac{\Delta}{a_1^2 a_2^2}\right)\right]\,. \tag{3.60}$$

Similarly the $m\omega$ equation (3.36) is now

$$\frac{2}{\Omega^2}\left(T_- K_-(r^2 - r_+^2) - T_+ K_+(r^2 - r_-^2)\right)$$
$$= \frac{-a_\star}{4(r_+^2 - r_-^2)}\left[-2(r^4 + \Delta r^2 Q_1^{01}) - 2\frac{a_1^2 a_2^2}{a_\star^2}(r^2 + \Delta r^2 Q_1^{02}) - 2(a_1^2 + a_2^2)\left(r^2 + \Delta r^2 Q_1^{11}\right)\right.$$
$$\left. -2\frac{a_1^2 a_2^2}{a_\star^2}(a_\star^2 + a_1^2 + a_2^2)\left(2 + \Delta r^2 Q_1^{12}\right) - 2\frac{a_1^4 a_2^4}{a_\star^2}\left(r^{-2} + \Delta r^2 Q_1^{22} - \frac{\Delta}{a_1^2 a_2^2}\right)\right]\,. \tag{3.61}$$

Together, matching the powers of $r$ in these equations sets values for $T_\pm$, $K_\pm$, which match the values found in [10, 16]. We also find two restrictions for the $Q_1^{ij}$:

$$Q_1^{11} = -2\frac{a_1^2 a_2^2}{a_\star^2}Q_1^{12} - \frac{a_1^4 a_2^4}{a_\star^4}Q_1^{22}\,,$$
$$Q_1^{01} = -1 - \frac{a_1^2 a_2^2}{a_\star^2}Q_1^{02} + \frac{a_1^4 a_2^4}{a_\star^4}Q_1^{12} + \frac{a_1^6 a_2^6}{a_\star^6}Q_1^{22}\,. \tag{3.62}$$

Thus there is a 3-parameter family of solutions. We will pick the simplest option, setting $Q_1^{02} = Q_1^{12} = Q_1^{22} = 0$. Accordingly, we have $Q_1^{11} = 0$ and $Q_1^{01} = Q_1^{10} = -1$, with all others set to zero.

Our proposed tensor equation thus becomes

$$-H_0^a H_0^b + \frac{1}{2}H_1^a H_{-1}^b + \frac{1}{2}H_{-1}^a H_1^b = \frac{1}{4}\left(r^2 g^{ab} + k_1^{ab} + l_0^a l_1^b + l_1^a l_0^b\right) - Q_1^{00} l_{(0)}^a l_{(0)}^b + E^{ab} + E_\star^{ab} \tag{3.63}$$

where $E_\star^{ab}$ is only allowed to have components along angular Killing vector directions other than $\phi_\star$. As before, this equation will be the same for $H$ and $\bar{H}$, and aside from the error term $E_\star$, independent of the choice of angle $\phi_\star$.

To characterize the error term $E^{ab}$, we begin by finding the cost to sandwich the $H$ generators, which is same for $\phi_\star = \phi_1$ or $\phi_\star = \phi_2$, as well as for $H$ or $\bar{H}$:

$$
\begin{aligned}
&-\left(\nabla_a H_0^a\right)H_0^b\nabla_b\Phi + \frac{1}{2}\left(\nabla_a H_1^a\right)H_{-1}^b\nabla_b\Phi + \frac{1}{2}\left(\nabla_a H_{-1}^a\right)H_1^b\nabla_b\Phi \\
&\qquad = \frac{a_1^2 a_2^2 + a_1^2 r^2 + a_2^2 r^2 - 2Mr^2 + r^4}{r(a_1^2 + a_2^2 + 2r^2 + (a_1^2 - a_2^2)\cos(2\theta))}\partial_r\Phi = \frac{1}{4}\nabla_a\left(r^2 g^{ab}\nabla_b\Phi\right).
\end{aligned}
\tag{3.64}
$$

As we found in the $4D$ case, (3.52) and (3.53), the cost for sandwiching $sr^{2(n-1-k)}$ exactly cancels this factor.[9] We thus see that the error term in the proposed tensor equation (3.63) becomes $\nabla_a(E^{ab} + E_\star^{ab})\nabla_b\Phi = \tilde{Q}^{00}\omega^2\Phi$ as before.

We will find it simpler to write both error terms together. For definiteness, we pick $\phi_\star = \phi_1$ below; a similar equation holds for the $\phi_\star = \phi_2$ case. The error term is independent of the choice of $H$ vs. $\bar{H}$. The error terms are

$$
\begin{aligned}
E^{ab} + E_1^{ab} = \frac{1}{4}\Bigg[&\left(-a_1^2 - a_2^2 + 2M + r^2\right)\delta_t^a\delta_t^b + \frac{a_2(a_1^2 + r^2)(a_1^2 + r_-^2)(a_1^2 + r_+^2)}{a_1^2(r^2 - r_-^2)(r^2 - r_+^2)}\left(\delta_t^a\delta_{\phi_2}^b + \delta_{\phi_2}^a\delta_t^b\right) \\
&\frac{a_2(a_1^2 + r_-^2)(a_1^2 + r_+^2)}{a_1(r^2 - r_-^2)(r^2 - r_+^2)}\left(\delta_{\phi_1}^a\delta_{\phi_2}^b + \delta_{\phi_2}^a\delta_{\phi_1}^b\right) + \frac{-a_1^4 + a_2^2 r^2 + a_1^2 a_2^2 + 2Ma_1^2 - a_1^2 r^2}{(r^2 - r_-^2)(r^2 - r_+^2)}\delta_{\phi_2}^a\delta_{\phi_2}^b\Bigg].
\end{aligned}
\tag{3.65}
$$

As expected, there are several error terms with $\delta_{\phi_2}^a$; these terms will always vanish when we turn off the momenta conjugate to $\phi_2$. The term proportional to $\delta_t^a\delta_t^b$, however, will only be eliminated when we take our near-region limit. Setting the constant $Q_1^{00} = a_1^2 + a_2^2 - 2M$ in (3.63), the tensor equation indicates that our near-region limit should obey

$$
r^2\omega^2 \to 0.
\tag{3.66}
$$

As in section 3.5 for $4D$, here we have now shown for $5D$ that matching the quadratic Casimir to the relevant pieces of the Klein-Gordon equation fixes $K_{L,R}$, $T_{L,R}$, and also builds a tensor equation whose error term defines our near region limit. For general dimensions, or for any $D \geq 4$ with a cosmological constant, we already needed to take a near-horizon limit in order to match the $r$-derivative pieces; this limit will itself make any error terms of the form (3.66) irrelevant compared to the near-horizon terms which are $\mathcal{O}(r - r_\pm)^{-1}$.

As we will see in the following section, the $K_{L,R}$, $T_{L,R}$ fixed here can also be found via a monodromy procedure (and indeed the $4D$ and $5D$ results match). Since the monodromy procedure itself studies the near-horizon behavior of the wave equation, we will actually be able to fix $K_{L,R}$, $T_{L,R}$ for general dimensions without taking any limits.

---

[9]The reader may wonder if the cost to sandwich the generators $H$ will always cancel the cost to sandwich the $sr^{2(n-1-k)}$ factor. Unfortunately we are not so lucky; the cancelation only holds in the $4D$ and $5D$ cases. Briefly, since the cost to sandwich $sr^{2(n-1-k)}$ depends on $k_j^{rr}$, it can in general depend on the $x_\mu$ coordinates for $\mu \neq n$; in other words, it can depend on the $\theta$-like directions. In specific, the general formula for this cost is $\partial_r(sU_n)g^{rr}\partial_r\Phi$, where the $U_n$ as defined in (2.28) depends on the other $\theta$-like directions $x_{\mu\neq n}$. In $4D$ and $5D$ where $n = 2$, there is only one other such coordinate, $s$ is a constant, and so $\partial_r(sU_n) = 2rs$. Conversely, for the $H$ generators, they only ever depend on $r$, so the cost to sandwich them will only be $r$ dependent.

# 4  Monodromy: $H_0$s for general $D$

In this section we use the monodromy approach [16–19] as reviewed in Section 2.1.2 to build the zero mode generators $H_0$ and $\bar{H}_0$ for an arbitrary dimension $D$. We also match our results for the $4D$ and $5D$ cases as worked out in [10, 15, 16].

We begin by computing the monodromies around the singular points $r_\pm$ of the radial wave equation (2.37). Expressing (2.37) in the standard form (2.15), we find

$$\lambda(r) = (r - r_\pm)\left(\frac{X_r'}{X_r} + \frac{\epsilon}{r}\right), \tag{4.1}$$

and

$$\gamma(r) = -(r - r_\pm)^2 \frac{\chi}{X_r^2}. \tag{4.2}$$

Here $\chi$ is again obtained after setting $\mu = n = r$ in (2.36),

$$\chi = X_r \sum_{j=0}^{n-1+\epsilon} K_j (r^2)^{n-1-j} - \left[\sum_{j=0}^{n-1+\epsilon} L_j (r^2)^{n-1-j}\right]^2, \tag{4.3}$$

and $X_r$ is given by[10]

$$X_r = 2Mr^{1-\epsilon} - [(r^2 + a_1^2)\dots(r^2 + a_{n-1+\epsilon}^2)](1 - \lambda r^2)r^{-2\epsilon}$$
$$= 2Mr^{1-\epsilon} - (-\lambda)^\sigma r^{-2\epsilon}\left(\sum_{j=0}^{N_{\epsilon,\sigma}} r^{2(N_{\epsilon,\sigma}-j)}\tilde{\mathcal{A}}^{(j)}\right). \tag{4.4}$$

Here we have again used the parameter $N_{\epsilon,\sigma}$ from (3.18), and $\tilde{\mathcal{A}}^{(k)}$ is defined as

$$\tilde{\mathcal{A}}^{(k)} = \sum_{\substack{i_1\dots i_k=1 \\ i_1 < \dots < i_k}}^{N_{\epsilon,\sigma}} a_{i_1}^2 \dots a_{i_k}^2, \tag{4.5}$$

where $\tilde{\mathcal{A}}^{(0)} = 1$ and $a_{N_{\epsilon,\sigma}}^2 = -1/\lambda$. Note that this definition matches $\mathcal{A}^{(k)}$ (2.27) when $\sigma = 0$. In (4.5), we have modified this definition to include the cosmological constant $\lambda$ in $\tilde{\mathcal{A}}^{(k)}$ so (A)dS backgrounds can be easily included in the calculations below.

Next, the parameters in the indicial equation (2.18) become

$$\lambda_0 = \lambda(r)|_{r=r_\pm} = 1, \tag{4.6}$$

and

$$\gamma_0^\pm = \gamma(r)|_{r=r_\pm} = -\frac{(r - r_\pm)^2 \chi}{X_r^2}\Big|_{r=r_\pm}. \tag{4.7}$$

---

[10]In order to take the $\lambda \to 0$ limit in (4.4), we need only set $\sigma = 0$, choosing of course $N_{\epsilon,\sigma} = N_{\epsilon,0} = n-1+\epsilon$ as well.

We can now solve the indicial equation (2.18) to find the monodromy parameters $\alpha_\pm$, finding

$$i\alpha_\pm = \frac{1 - \lambda_0 + \sqrt{(1 - \lambda_0)^2 - 4\gamma_0^\pm}}{2} = i\sqrt{\gamma_0^\pm}. \tag{4.8}$$

In order to simplify the evaluation of $\gamma_0$ in (4.7) near the horizons[11] $r_\pm$, we factor out $r - r_\pm$ from $X_r$, writing

$$X_r = (r - r_\pm)\mathcal{P}^\pm(r), \tag{4.9}$$

where $r^{2\epsilon}\mathcal{P}^\pm(r)$ is a polynomial with terms $r^0$ up to $r^{2N_{\epsilon,\sigma}-1}$. Now, matching the coefficients of all powers of $r$ in (4.4) and (4.9) we find

$$\mathcal{P}^\pm(r) = r^{-2\epsilon}(-\lambda)^\sigma \left[ \frac{\tilde{\mathcal{A}}^{(N_{\epsilon,\sigma})}}{r_\pm} - \sum_{k=0}^{N_{\epsilon,\sigma}-1} \tilde{\mathcal{A}}^{(k)} \sum_{j=1}^{2(N_{\epsilon,\sigma}-k)-1} r^j r_\pm^{2(N_{\epsilon,\sigma}-k)-1-j} + \epsilon r \left( \sum_{j=0}^{N_{\epsilon,\sigma}} r_\pm^{2(N_{\epsilon,\sigma}-j-1)} \tilde{\mathcal{A}}^{(j)} \right) \right]. \tag{4.10}$$

Now evaluating $\mathcal{P}^\pm(r)$ at $r_\pm$ gives

$$\mathcal{P}^\pm(r_\pm) = r_\pm^{-2\epsilon}(-\lambda)^\sigma \sum_{k=0}^{N_{\epsilon,\sigma}} (1 + \epsilon - 2k)\tilde{\mathcal{A}}^{(N_{\epsilon,\sigma}-k)} r_\pm^{2k-1}. \tag{4.11}$$

Using (4.7) and (4.8) we find that the monodromy parameters around the singular points $r_\pm$ are

$$\alpha_\pm = i\frac{\sqrt{\chi(r_\pm)}}{\mathcal{P}^\pm(r_\pm)}. \tag{4.12}$$

The result (4.12) suggests a connection between the monodromies and the Killing tower. As we point out in Section 3.1 and Appendix B, the horizon entropy is related to the Noether charge associated with the Killing field that vanishes on the black hole bifurcation surface, which in turn is related to monodromies [16]. The entire Killing tower structure should also have a similar thermodynamic interpretation, as we will explore further in the discussion in section 6.

Now that we have the monodromies built we can compute the energy eigenvalues (2.20) associated with the two Killing horizons,

$$\begin{aligned} \omega_L &= \frac{i\sqrt{\chi(r_+)}}{\mathcal{P}^+(r_+)} - \frac{i\sqrt{\chi(r_-)}}{\mathcal{P}^-(r_-)}, \\ \omega_R &= \frac{i\sqrt{\chi(r_+)}}{\mathcal{P}^+(r_+)} + \frac{i\sqrt{\chi(r_-)}}{\mathcal{P}^-(r_-)}, \end{aligned} \tag{4.13}$$

---

[11]The roots of $X_r$ are the regular singular points. Even for arbitrary dimension $D$, for flat space and AdS, it turns out there are only two real positive (physical) roots $r_\pm$ for such black holes. For de Sitter black holes there are also real roots for the cosmological horizons.

where

$$\sqrt{\chi(r_\pm)} = i \sum_{j=0}^{n-1+\epsilon} L_j r_\pm^{2(n-1-j)} \, . \tag{4.14}$$

As in sections 3.1 and 3.3, we now choose the Boyer-Lindquist time $t$ and a specific Boyer-Lindquist angle $\phi_\star$, where we have denoted the chosen $\mu$ index via $\star$. We then expand the Klein-Gordon scalar in Fourier modes, $\Phi \propto e^{im\phi_\star - i\omega t}$, and find the $L_i$ as in (3.30). With the $L_i$, we find

$$i\sqrt{\chi(r)} = \sum_{k=0}^{n+\epsilon-1} (-ma_\star(\lambda\mathcal{A}_\star^{(k)} - \mathcal{A}_\star^{(k-1)}) + \omega\mathcal{A}^{(k)})r^{2(n-1-k)} \, . \tag{4.15}$$

We note that this equation uses $\mathcal{A}^{(k)}$ from (2.27) and (3.31). We can now identify the $\omega$ and $m$ components of $\omega_{L,R}$ (4.13). We write

$$\omega_R = A\omega - Bm, \tag{4.16}$$

$$\omega_L = C\omega - Dm, \tag{4.17}$$

and identify

$$A = \sum_{k=0}^{n+\epsilon-1} \mathcal{A}^{(k)} \left[ \frac{r_+^{2(n-k-1)}}{\mathcal{P}^+(r_+)} + \frac{r_-^{2(n-k-1)}}{\mathcal{P}^-(r_-)} \right] , \tag{4.18a}$$

$$B = a_\star \left[ \sum_{k=0}^{n+\epsilon-1} \left( \lambda\mathcal{A}_\star^{(k)} - \mathcal{A}_\star^{(k-1)} \right) \left( \frac{r_+^{2(n-k-1)}}{\mathcal{P}^+(r_+)} + \frac{r_-^{2(n-k-1)}}{\mathcal{P}^-(r_-)} \right) \right] , \tag{4.18b}$$

$$C = \sum_{k=0}^{n+\epsilon-1} \mathcal{A}^{(k)} \left[ \frac{r_+^{2(n-k-1)}}{\mathcal{P}^+(r_+)} - \frac{r_-^{2(n-k-1)}}{\mathcal{P}^-(r_-)} \right] , \tag{4.18c}$$

$$D = a_\star \left[ \sum_{k=0}^{n+\epsilon-1} \left( \lambda\mathcal{A}_\star^{(k)} - \mathcal{A}_\star^{(k-1)} \right) \left( \frac{r_+^{2(n-k-1)}}{\mathcal{P}^+(r_+)} - \frac{r_-^{2(n-k-1)}}{\mathcal{P}^-(r_-)} \right) \right] . \tag{4.18d}$$

As we will show below, the zero mode generators, as well as the exponents of the conformal coordinates, can be found in terms of these parameters.

Next, we find $t_R$ and $t_L$ such that

$$e^{-i\omega_R t_R - i\omega_L t_L} = e^{-i\omega t + im\phi}. \tag{4.19}$$

Since $t_R$, $t_L$, $t$, $\phi$ are all spacetime coordinates and $A$, $B$, $C$, $D$ are all dependent on only black hole background parameters, the only way to satisfy this equation is to match coefficients of $\omega$ and $m$ on both sides. From this matching we find the matrix equation

$$\begin{bmatrix} t \\ \phi \end{bmatrix} = \begin{bmatrix} A & C \\ B & D \end{bmatrix} \begin{bmatrix} t_R \\ t_L \end{bmatrix} . \tag{4.20}$$

We can now identify the zero mode generators as

$$\begin{aligned} H_0 &= i\partial_{t_R} = Ai\partial_t + Bi\partial_\phi \, , \\ \bar{H}_0 &= -i\partial_{t_L} = -Ci\partial_t - Di\partial_\phi \, . \end{aligned} \tag{4.21}$$

Additionally, $t_R$, $t_L$, which appear in the exponents of the conformal coordinates, can be solved for by inverting the matrix equation (4.20).

Our results above are generic for Kerr-(A)dS black holes with arbitrary dimension, spin, and cosmological constants. Although we have not done so here, adding NUT charges should be straightforward.

Instead, we now show that our results are consistent with previous results for the $H_0$ and $\bar{H}_0$ in $4D$ flat, $5D$ flat, and $4D$ AdS black holes with arbitrary spins. Beginning with the $4D$ asymptotically flat black hole, we find

$$\mathcal{P}_\pm^{4D} = \frac{(a^2 - r_\pm^2)}{r_\pm},\tag{4.22}$$

which leads to the zero mode generators

$$H_0^{4D} = -2iM\partial_t, \qquad \bar{H}_0^{4D} = \frac{2ia}{r_+ - r_-}\partial_\phi + \frac{2iM(r_+ + r_-)}{r_+ - r_-}\partial_t.\tag{4.23}$$

This result matches with [8], as well as to our rederivation as found in section 3.5.

For the $5D$ asymptotically flat case, we have

$$\mathcal{P}_\pm^{5D} = \frac{2(a_1^2 a_2^2 - r_\pm^4)}{r_\pm^3} = \mp\frac{2(r_+^2 - r_-^2)}{r_\pm}.\tag{4.24}$$

We then specify the choice $\phi_\star = \phi_1$, which, along with the useful facts (3.59), leads to the zero mode generators

$$H_0^{5D} = \frac{-iM}{(r_+ + r_-)}\partial_t + \frac{i(a_1 - a_2)}{2(r_+ + r_-)}\partial_{\phi_1}, \qquad \bar{H}_0^{5D} = \frac{iM}{(r_+ - r_-)}\partial_t - \frac{i(a_1 + a_2)}{2(r_+ - r_-)}\partial_{\phi_1}.\tag{4.25}$$

These generators match [10, 16][12]. Additionally these generators match those rederived for $5D$ in section 3.6. To obtain the generators for $\phi_2$ instead, simply exchange $1 \leftrightarrow 2$.

For the case of the (A)dS asymptotics in $4D$, we find

$$\mathcal{P}_\pm^{4D} = \frac{a^2 + (\lambda a^2 - 1)r_\pm^2 + 3\lambda r_\pm^4}{r_\pm}.\tag{4.26}$$

In this case the zero mode generators become

$$H_0^{4D,\Lambda} = \left(\frac{r_+^3 + a^2 r_+}{a^2 + (\lambda a^2 - 1)r_+^2 + 3\lambda r_+^4} + \frac{r_-^3 + a^2 r_-}{a^2 + (\lambda a^2 - 1)r_-^2 + 3\lambda r_-^4}\right)i\partial_t \tag{4.27}$$
$$+ \left(\frac{a\lambda r_+^3 - ar_+}{a^2 + (\lambda a^2 - 1)r_+^2 + 3\lambda r_+^4} + \frac{a\lambda r_-^3 - ar_-}{a^2 + (\lambda a^2 - 1)r_-^2 + 3\lambda r_-^4}\right)i\partial_\phi,$$

$$\bar{H}_0^{4D,\Lambda} = -\left(\frac{r_+^3 + a^2 r_+}{a^2 + (\lambda a^2 - 1)r_+^2 + 3\lambda r_+^4} - \frac{r_-^3 + a^2 r_-}{a^2 + (\lambda a^2 - 1)r_-^2 + 3\lambda r_-^4}\right)i\partial_t \tag{4.28}$$
$$- \left(\frac{a\lambda r_+^3 - ar_+}{a^2 + (\lambda a^2 - 1)r_+^2 + 3\lambda r_+^4} - \frac{a\lambda r_-^3 - ar_-}{a^2 + (\lambda a^2 - 1)r_-^2 + 3\lambda r_-^4}\right)i\partial_\phi.$$

---

[12]Convention differences include the sign of angles and the choice of how to assign left and right movers to the sum or difference of $\alpha_\pm$.

Our results here are not easily matchable to the basis chosen in [15], but they do recover the $\lambda \to 0$ results in (4.23).[13] In the next section, we will analyze the Schwarzschild black hole in the large $D$ limit, and will again recover monodromy results compatible with our general dimension, general spin results in (4.18a).

# 5 A worked example: The Schwarzschild Black Hole in Two Large $D$ Limits

Thus far, we have connected the hidden conformal symmetry from Kerr/CFT directly to the hidden Killing tower symmetries of rotating black hole spacetimes, via the tensor equations built in sections 3.5 and 3.6. Then, we have confirmed the zero mode generators $H_0$ and $\bar{H}_0$ via the monodromy approach in 4.

Since the calculations in 4 apply to all Kerr-(A)dS black holes in general dimensions, and the same spacetimes also possess a full Killing tower, we expect a similar tensor equation can be found, in an appropriate limit, for all of these spacetimes. We will now show that exactly this connection can be built for the large $D$ Schwarzschild black hole.

We begin with a brief review of the large dimension limit (for a recent full review, see [35]). As first shown in [25], taking the large dimension (or large $D$) limit of the Schwarzschild black hole results in two important regions: a flat 'far' region whose fluctuations do not see the black hole, and a near-horizon 'membrane' region which encodes the most important black hole physics [29, 30]. We can explicitly see these regions by considering the metric for a Schwarzschild black hole of radius $r_0$ in general dimensions:

$$ds^2 = -f(r)dt^2 + \frac{dr^2}{f(r)} + r^2 d\Omega_{D-2}, \qquad f(r) = 1 - \left(\frac{r_0}{r}\right)^{D-3}. \tag{5.1}$$

The flat region appears if we fix any $r > r_0$ and let $D \to \infty$. However, if we instead examine a near-horizon region by setting $r = r_0 \left(1 + \frac{\lambda}{D-3}\right)$ and fixing $\lambda$ as we take $D \to \infty$, we obtain the 'membrane' region. Using the radial coordinate $\rho = (r/r_0)^{D-3}$, and setting $\bar{n} = D - 3$ the metric in this region becomes

$$ds^2 = -\left(1 - \frac{1}{\rho}\right)dt^2 + \frac{r_0^2}{\bar{n}^2}\frac{d\rho^2}{\rho(\rho - 1)} + r_0^2 d\Omega_{\bar{n}+1}. \tag{5.2}$$

As two of us showed in [34], the off-shell graviton quasinormal modes in this background are given by hypergeometric functions with integer parameters, indicative of an underlying $SL(2, R)$. We will now make this connection explicit, by building towards a set of conformal coordinates whose $H$ generators have a Casimir that matches the Klein-Gordon equation, and then showing the zero mode generators match the monodromy analysis.

Afterwards, we consider instead beginning with the metric (5.1), building the scalar Klein-Gordon equation, and then taking a large $D$ limit. We find this approach still requires

---

[13]To show that (4.27) becomes (4.23) in the $\lambda \to 0$ limit, first set $\lambda = 0$ explicitly. Then, use the root relationship, valid only at $\lambda = 0$, that $r_+ r_- = a^2$. The general $\lambda$ version of this relationship is $\left[r_+ r_- - \frac{a^2}{\lambda r_+ r_-} - (r_+ r_-)^2\right] = a^2 - \frac{1}{\lambda}$, and the roots also solve $a^2 r_- + a^2 r_+ + (r_+ + r_-)\lambda r_+^2 r_-^2 = 2M r_+ r_-$.

an explicit near-horizon limit in order to match the $R$ coefficient terms in the Klein-Gordon equation. A monodromy analysis provides this limit naturally and thus recovers the results from the metric limit.

## 5.1 The metric Large $D$ Limit

In this section, we consider the large $D$ limit taken in the metric. Specifically, we will start with the metric (5.2), and build the scalar Klein-Gordon equation:

$$(\rho - 1)\partial_\rho^2 \Phi + \partial_\rho \Phi + \left[ \frac{-\ell(\ell + \bar{n})}{\bar{n}^2 \rho} + \frac{\omega^2}{\bar{n}^2(\rho - 1)} \right] \Phi = 0. \tag{5.3}$$

Here, $\bar{n} = D - 3$ as above, $\ell$ is the total angular momentum from the $\bar{n} + 1$ angular directions, and $\omega$ is the frequency conjugate to the static coordinate $t$. We have additionally set $r_0 = 1$ for simplicity.

The next step is to find conformal coordinates, of the form (3.1), whose $H$ generators have a Casimir that matches (5.3). Since again the radially separated equation here has no $\partial_\rho \partial_t$ or $\partial_\rho \partial_\psi$ crossterms (for any angle $\psi$), the radial functions $g(\rho)$ and $h(\rho)$ must still satisfy (3.9). Accordingly, their Casimir must be of the form (3.10).

We begin by matching the ratio of the $\partial_\rho^2$ and $\partial_\rho$ terms between the Klein-Gordon equation (5.3) and the Casimir (3.10). This equation looks like the general Klein-Gordon equation (2.37), if we use $\rho$ as the radial coordinate instead of $r$ and set $\Delta = (\rho - 1)$ with $\epsilon = 0$. Thus we require

$$\rho - 1 = \frac{c_1 h(1 + h^2)}{\partial_\rho h}, \tag{5.4}$$

which is solved by $h^2 = e^I/(1 - e^I)$ as in (3.15). Here we find

$$e^I = \tilde{c}_2 (\rho - 1)^{2c_1}, \tag{5.5}$$

where $c_1$ and $\tilde{c}_2 = e^{c_2}$ are (as yet unfixed) constants.

In order to match both radial terms, we must multiply the Klein-Gordon equation (5.3) by an overall factor just as in (3.12), which we again term $s$. Matching to the radial second derivative term in the Casimir (3.10), or using (3.17), we find

$$s = \frac{(\rho - 1)\left(1 - e^I\right)^2}{4c_1^2 e^I} = \frac{\left[1 - \tilde{c}_2(\rho - 1)^{2c_1}\right]^2}{4c_1^2 \tilde{c}_2 (\rho - 1)^{2c_1 - 1}}. \tag{5.6}$$

We begin by ensuring the $\partial_\psi \partial_t$ cross term in (3.10) vanishes, since there is no cross term in our large $D$ Klein-Gordon equation. We require

$$\frac{T_- K_-}{h^2 + 1} = \frac{T_+ K_+}{h^2}. \tag{5.7}$$

Since the radial dependence of these two terms is different, both sides must vanish independently. That is, we require $T_- K_- = T_+ K_+ = 0$.

Next, we match the $\partial_\psi^2$ term in (3.10) to the $\ell$-dependent term in the Klein-Gordon equation (5.3). Since we expect to turn on the momentum in only one angular direction, we will identify $\ell(\ell + \bar{n})$ with the momentum in the $\psi$ direction:

$$\frac{K_-^2}{\Omega^2(h^2+1)} - \frac{K_+^2}{\Omega^2 h^2} = \frac{-s}{\bar{n}^2 \rho}. \tag{5.8}$$

Rewriting both sides in terms of $e^I$, we find

$$\frac{K_-^2\left(1-e^I\right)}{\Omega^2} - \frac{K_+^2\left(1-e^I\right)}{\Omega^2 e^I} = -\frac{(\rho-1)\left(1-e^I\right)^2}{\bar{n}^2 4 c_1^2 e^I \rho} \tag{5.9}$$

Solving for $e^I$ we obtain

$$e^I = \frac{4 c_1^2 \bar{n}^2 K_+^2 \rho - \Omega^2 \rho + \Omega^2}{4 c_1^2 \bar{n}^2 K_-^2 \rho - \Omega^2 \rho + \Omega^2} = \tilde{c}_2(\rho-1)^{2c_1}, \tag{5.10}$$

where in the last equality we have plugged in our solution for $e^I$ (5.5). This equation and the $m\omega$ equation (5.7) are solved simultaneously by setting

$$\tilde{c}_2 = -1,\ c_1 = \frac{1}{2},\ K_+ = 0,\ T_- = 0,\ T_+ = \frac{\bar{n}}{2\pi}. \tag{5.11}$$

From these parameters we find

$$e^I = 1 - \rho,\ s = -\rho^2. \tag{5.12}$$

If we set our last remaining parameter to $K_- = \bar{n}/2\pi$, then we find the leftover term is

$$\frac{\omega^2 \rho}{\bar{n}^2}. \tag{5.13}$$

As expected this term does not have a pole at $\rho = 1$; thus it is not important for the monodromy behavior around the horizon at $\rho = 1$, and we can argue that it will be removed if we study either a near-horizon or near-region limit. Formally we would then write

$$K_L = K_R = \frac{\bar{n}}{4},\ T_L = T_R = \frac{\bar{n}}{4\pi}, \tag{5.14}$$

but the matching between the two temperatures here is actually somewhat specious; rather than a full CFT, we expect only a chiral CFT with only one temperature.

Studying the monodromy behavior of the Klein-Gordon equation (5.3), we first find $\lambda$ and $\gamma$ from writing the equation in the standard form (2.15):

$$\lambda(\rho) = 1,\qquad \gamma(\rho) = \left[\frac{-\ell(\ell+\bar{n})(\rho-1)}{\bar{n}^2\rho} + \frac{\omega^2}{\bar{n}^2}\right]. \tag{5.15}$$

Evaluating at $\rho = 1$ and solving the indicial equation (2.18), we find

$$\alpha_1 = \frac{\omega}{\bar{n}}. \tag{5.16}$$

The equation does not have a second singular point; rather the singular point at $\rho = 1$ is repeated. Accordingly, just as matching the Casimir directly, we only see evidence of a single chiral theory with one temperature.

## 5.2    The Large $D$ Limit in the Klein-Gordon Equation

In this section, we instead study the exact separated Klein-Gordon equation, and only work out where the large $D$ limit is needed when we are matching the Casimir to the field equation itself. We will start building towards a quadratic Casimir, but will not succeed at matching the $R$ coefficients, only being able to match the $R'$ and $R''$, at least if we do not further expand around the horizon. The situation for this exact Schwarzschild metric in general dimensions is thus akin to the behavior for the more general class of black holes we considered previously; either a monodromy approach which naturally studies the near-horizon behavior, or an explicit near-horizon limit, must be taken.

We now consider the Klein-Gordon equation in the general dimension Schwarzschild metric (5.1) written in the coordinate $\rho = (r/r_0)^{\bar{n}}$ but without taking a large $\bar{n}$ limit. We find

$$\rho(\rho - 1)\bar{n}^2\partial_\rho^2\Phi + \bar{n}^2 (2\rho - 1) \partial_\rho\Phi + \left(-\ell(\ell + \bar{n}) + \frac{\rho^{1+2/\bar{n}}\omega^2}{\rho - 1}\right) \Phi = 0 \,. \qquad (5.17)$$

Here, we can use $\Delta = \rho(\rho - 1)$ as in the previous section, and find again that $h^2 = e^I/(1 - e^I)$ as in (3.15), with

$$e^I = \tilde{c}_2 \left(\frac{\rho - 1}{\rho}\right)^{2c_1} \,, \qquad (5.18)$$

for as yet unfixed constants $\tilde{c}_2$ and $c_1$. As in the previous section, we must have $T_-K_- = T_+K_+ = 0$ since $m\omega$ terms are not present in the scalar equation (5.17). We also find a similar equation for matching the $m^2$ pieces; after plugging in our new value for $e^I$ we find

$$\tilde{c}_2 \left(\frac{\rho - 1}{\rho}\right)^{2c_1} = \frac{4c_1^2K_+^2 - \Omega^2\bar{n}^2\rho(\rho - 1)}{4c_1^2K_-^2 - \Omega^2\bar{n}^2\rho(\rho - 1)} \,. \qquad (5.19)$$

After some algebraic rearrangement, we find

$$4c_1^2\tilde{c}_2 K_-^2 (\rho - 1)^{2c_1} - \tilde{c}_2\Omega^2\bar{n}^2\rho(\rho - 1)^{2c_1+1} = 4c_1^2K_+^2\rho^{2c_1} - \Omega^2\bar{n}^2\rho^{2c_1+1}(\rho - 1) \,. \qquad (5.20)$$

Now we can clearly see the problem. Matching the leading $\rho^{2c_1+2}$ behavior fixes $\tilde{c}_2 = 1$, since only the second term on each side of the equation is involved. However, the next power down, $\rho^{2c_1+1}$, then cannot match; again only the second term on each side is involved, but the coefficient on one side has a factor of $2c_1 + 1$ from the binomial expansion. Accordingly, even though we were able to match the $\partial_\rho^2$ and $\partial_\rho$ terms, and even though we could pick $e^I$ to be a ratio of polynomials, we cannot proceed to match this $m^2$ term in the non-derivative piece without a further near-horizon limit.

Rather than attempt to take this limit explicitly here, we will instead study the monodromy approach. Rewriting the equation (5.17) in standard form around $\rho = 1$, we find

$$\lambda_0 = \left.\frac{2\rho - 1}{\rho}\right|_{\rho=1} = 1 \,, \qquad \gamma_0 = \left.\left(\frac{-\ell(\ell + n)(\rho - 1)}{\rho\bar{n}^2} + \frac{\rho^{2/n}\omega^2}{n^2}\right)\right|_{\rho=1} = \frac{\omega^2}{n^2} \,. \qquad (5.21)$$

Using the indicial equation (2.18), we again find the monodromy parameter

$$\alpha_1 = \frac{\omega}{n}.$$

(5.22)

Additionally, as before, the only horizon is at $\rho = 1$; this single monodromy parameter is thus our only information about the near-horizon behavior necessary to set the temperature of the chiral CFT. Unsurprisingly, this result exactly matches the metric limit parameter, (5.16), because the explicit near-horizon metric used there already focuses on exactly the same information the monodromy method does given its focus on behavior near the singular points.

In order to do a more full large $D$ analysis, we would want to allow for inner and outer horizons; the simplest case to study would be the Myers-Perry black hole with all spins $a_i$ set equal to a single value $a$. We leave this analysis to future work.

## 6   Discussion

In this work, we have constructed elements of the hidden conformal symmetry narrative of [8] and [18] directly from the Killing tower objects that guarantee separability of the wave equation. In $4D$ and $5D$, we built a tensor equation for the quadratic Casimir $\mathcal{H}^2$ of [8] and [10]. We then built the monodromy parameters $\alpha_\pm$ of [16–19] directly from the Killing tower for Kerr-(A)dS black holes in general dimension. We also used this machinery to calculate the hidden conformal symmetry generators for large $D$ Kerr-(A)dS black holes. We hope that this will be a step toward establishing a Large-$D$/CFT correspondence, since many have shown [35] that much of the key black hole physics is captured by the large dimension limit.

We believe that there are many avenues for future work in this program. For example, one important aspect of the generators (2.6) proposed by [8] is that they are not globally defined. That is, they are not invariant under the identification $\phi \sim \phi + 2\pi$, and the $SL(2, R) \times SL(2, R)$ symmetry is spontaneously broken to $SL(2, R) \times U(1)$. Recently, a different set of *globally defined* symmetry generators of the near-region Klein-Gordon equation were defined in [43]. In addition, the generators of [43] possess a smooth Schwarzschild limit $a \to 0$, and reproduce the hidden symmetry generators for Schwarzschild found in [44]. It would be interesting to analyze what role these globally defined generators might play in the hidden conformal symmetry narrative outlined in this paper. For example, why does the monodromy method seem to naturally reconstruct the locally defined generators (2.6) instead of the globally defined ones of [43]?

It would be interesting if this framework could be used to provide a thermodynamic description of the Killing tower objects of separability and integrability. Our equation for the monodromies, (4.12), is essentially a relationship between 1) the conserved quantities $L$ from the Killing tower (2.33) and 2) the conserved charges $\alpha_\pm$ associated with Wald's Killing vectors (3.3) that vanish on the inner and outer horizons.[14] We leave further development of this relationship for future work.

---

[14]Further discussion of $\alpha_\pm$ as conserved charges can be found in [16] and Appendix B.

In the $4D$ and $5D$ asymptotically flat black holes, early work [8–10] studying the hidden conformal symmetry needed a 'near-region' limit to remove the mismatch between the Casimir and the full radial Klein-Gordon operator. However, subsequent work [18] has demonstrated that this near-region limit is not necessary to probe the hidden conformal symmetry at the horizon. Indeed, the monodromy method teaches us that only the lowest order contribution in an expansion around either the inner or outer horizon is necessary to reproduce the $SL(2, R)$ generators found by [8] and [10].

Conversely, in this work, for $D \geq 6$ in asymptotically flat space and for $D \geq 4$ with a cosmological constant, we see that a near-horizon limit (by which we mean keeping only terms at leading order near $r = r_{\pm}$, regardless of the relative value of $\omega$) is necessary in our Killing tensor construction as well. In Section 3.2, we needed this limit in order to match the quadratic Casimir to the r-derivative terms in the Klein-Gordon equation. Since this matching is the first step in the procedure to build the conformal coordinates, taking this limit is necessary before matching the Killing vector directions, which fix the $K_{L,R}$ and $T_{L,R}$ parameters.

However, in the monodromy approach, since it intrinsically captures the behavior near singular points of the wave equation, no separate near-horizon limit is needed to find the $K_{L,R}$ and $T_{L,R}$ in any dimension. Consequently, we were able to explicitly find these parameters, which fix the $t$ and $\phi$ dependence in the conformal coordinates, without requiring a near-horizon limit; our expressions are valid for the general spin Kerr-(A)dS black hole. Of course, if we want the conformal coordinates to actually reproduce the wave equation, we still need to match the $r$-derivative pieces which would then force an explicit near-horizon limit.

In future work, we will work towards building a tensor equation for general dimensions. We believe the best approach will be to combine our general dimension radial coordinate result in (3.15) and (3.23), with our general dimension monodromy result (4.18a). The radial coordinate result sets the radial dependence of the conformal coordinates, but its poor analytic behavior requires focusing on near-horizon information. Rather than take an explicit near-horizon limit, it is logical to use the monodromy approach to fix the $t$ and $\phi$ dependence of the conformal coordinates, since it naturally focuses on the analytic behavior near the inner and outer horizon already. Unfortunately this combined approach does not allow immediate use of the Klein-Gordon equation written in terms of the Killing tensors $k_{(j)}^{ab}$, so we leave its study to future work.

Another important point regarding our tensor equations for $4D$ and $5D$, (3.51) and (3.63), and possible future generalizations to higher dimensions, is how they rewrite the inverse metric as sum of 'squared' terms (up to the near-region limit). The quadratic Casimir itself is a sum of paired products of $H$ generators, while the remaining terms are either products of two Killing vectors, or the single Killing tensor term $k_1^{ab}$. However even the single Killing tensor term can itself be written as $k_1^{ab} = f_c^a f^{bc}$, where $f$ is the Killing-Yano tensor. This clear "squared" rewriting hints at a classical double-copy picture for the generic Kerr-NUT-AdS black holes, since the writing in terms of Killing-Yano tensors is possible (although more complicated) for general dimensions. Again we leave this connection to future work.

Lastly, since diagnosing hidden conformal symmetry requires studying the dynamics of a probe field on a black hole background, it is interesting to ask the question: does changing the dynamics affect the presence of hidden conformal symmetry? In a forthcoming work [45], we investigate hidden conformal symmetry in higher derivative theories of gravity, so that the equation of motion is $(\nabla^\mu \nabla_\mu)^r \Phi = 0$, for general integer $r > 1$. This theory is additionally interesting to look at, since known holographic duals to logarithmic conformal field theories (Log CFTs) involve higher derivative interactions [46]. Finding hidden conformal symmetry in this setting could thus point to a new instance of a Log CFT correspondence. It is perhaps especially interesting to note that this higher derivative theory is nonunitary, and so this could be an example of a Cardy formula reproducing the Bekenstein-Hawking entropy of a black hole in a nonunitary situation.

## Acknowledgements

We thank Alex Chanson, Fridrik Freyr Gautason, Valentina Giangreco M. Puletti, Rahul Poddar, Maria Rodriguez, Watse Sybesma and Lárus Thorlacius for illuminating conversations. The work of CK and AP is supported by the U.S. Department of Energy under grant number DE-SC0019470. The work of VM is supported by the Icelandic Research Fund under grant 195970-052.

## A    Metrics and notation

The Kerr metric is Boyer-Lindquist coordinates is

$$ds^2 = \frac{\rho^2}{\Delta}dr^2 - \frac{\Delta}{\rho^2}(dt^2 - a\,\sin^2\theta d\phi)^2 + \rho^2 d\theta^2 + \frac{\sin^2\theta}{\rho^2}((r^2 + a^2)d\phi - adt)^2, \qquad (A.1)$$

with definitions $\Delta = r^2 + a^2 - 2Mr$ and $\rho^2 = r^2 + a^2\cos^2\theta$. The spin parameter $a$ is the ratio of the black hole's angular momentum $J$ and mass $M$: $a \equiv \frac{J}{M}$. The surface gravities and angular velocities associated with the inner and outer horizons $r_\pm = M \pm \sqrt{M^2 - a^2}$ are

$$\kappa_\pm = \frac{r_+ - r_-}{4Mr_\pm}, \qquad \Omega_\pm = \frac{a}{2Mr_\pm}, \qquad (A.2)$$

respectively. The Kerr black hole has two temperatures related to the inner and outer horizons. They are commonly written as

$$T_L = \frac{r_+ + r_-}{4\pi a}, \qquad T_R = \frac{r_+ - r_-}{4\pi a}. \qquad (A.3)$$

The general dimension Kerr-(A)dS metric in Boyer-Lindquist-like coordinates is given by [21, 47, 48]

$$
\begin{aligned}
g = & -\left(1 - \lambda r^2\right) W dt^2 \\
& + \frac{2Mr^{1-\epsilon}}{\Sigma} \left( W \, dt + \sum_{\nu=1}^{n-1+\epsilon} \frac{\mu_\nu^2 a_\nu}{1 + \lambda a_\nu^2} \, d\phi_\nu \right)^2 \\
& + \frac{\Sigma}{\Delta} dr^2 + (1 - \epsilon) r^2 d\mu_0^2 + \sum_{\nu=1}^{n-1+\epsilon} \frac{r^2 + a_\nu^2}{1 + \lambda a_\nu^2} \left( d\mu_\nu^2 + \mu_n^2 d\phi_\nu^2 \right) \\
& + \frac{\lambda}{(1 - \lambda r^2) W} \left( (1 - \epsilon) r^2 \mu_0 d\mu_0 + \sum_{\nu=1}^{n-1+\epsilon} \frac{r^2 + a_\nu^2}{1 + \lambda a_\nu^2} \mu_\nu d\mu_\nu \right)^2 .
\end{aligned}
\tag{A.4}
$$

Here the metric functions are given by

$$
\begin{aligned}
\Delta &= \left(1 - \lambda r^2\right) \prod_{\nu=1}^{n-1+\epsilon} \left(r^2 + a_\nu^2\right) r^{-2\epsilon} - 2Mr^{1-\epsilon} , \\
\Sigma &= \left( (1 - \epsilon) \mu_0^2 + \sum_{\nu=1}^{n-1+\epsilon} \frac{r^{2-2\epsilon} \mu_\nu^2}{r^2 + a_\nu^2} \right) \prod_{\mu=1}^{n-1+\epsilon} \left(r^2 + a_\mu^2\right) , \\
W &= (1 - \epsilon) \mu_0^2 + \sum_{\nu=1}^{n-1+\epsilon} \frac{\mu_\nu^2}{1 + \lambda a_\nu} ,
\end{aligned}
\tag{A.5}
$$

and

$$
\sum_{\nu=\epsilon}^{n-1+\epsilon} \mu_\nu^2 = 1 .
\tag{A.6}
$$

The dimensionality of the metric is given by $D = 2n + \epsilon$, where $\epsilon = 0$ for even dimensions and $\epsilon = 1$ for odd dimensions. The parameter $a_\nu$ is the spin associated with the angle $\phi_\nu$.

## B    The $(\omega_L, \omega_R)$ basis

In this appendix, we discuss the change of basis

$$
\omega_L = \alpha_+ - \alpha_-, \qquad \omega_R = \alpha_+ + \alpha_-.
\tag{B.1}
$$

We first explore this from a thermodynamic point of view. As was mentioned in [16], the monodromy parameters $\alpha_\pm$ are related to Wald's interpretation of black hole entropy as a Noether charge [42]. That is, consider a stationary black hole with bifurcate Killing horizon. There is a particular Killiing field that vanishes on the bifurcation surface $\Sigma$. For Kerr, it is

$$
\zeta^\pm = (\partial_t + \Omega_\pm \partial_\phi),
\tag{B.2}
$$

where $\Omega_\pm$ are defined in (A.2), and (B.2) is normalized to have unit surface gravity $\kappa_\pm$. Then the horizon entropy $S_\pm$ is $2\pi$ times the integral of the Noether charge associated with (B.2) over the bifurcation surface $\Sigma$.

We can also see that the choice (B.1) gives the Wald generators (B.2) a CFT interpretation. Consider again our change of basis

$$e^{-i\omega_L t_L - i\omega_R t_R} = e^{-i\omega t + im\phi}. \tag{B.3}$$

With the choice (B.1), the conjugate variables

$$t_R = 2\pi T_R \phi, \qquad t_L = \frac{1}{2M} t - 2\pi T_L \phi. \tag{B.4}$$

are related to the zero mode generators $(H_0, \bar{H}_0)$ by

$$H_0 = \frac{i}{2\pi T_R} \partial_\phi + 2iM \frac{T_L}{T_R} \partial_t = i\partial_{t_R}, \qquad \bar{H}_0 = -2iM\partial_t = -i\partial_{t_L}. \tag{B.5}$$

Taking these seriously as CFT generators, we can define a Hamiltonian $H = H_0 + \bar{H}_0$ and angular momentum $J = H_0 - \bar{H}_0$ in the usual way. We find

$$
\begin{aligned}
H &= H_0 + \bar{H}_0 = i(\partial_{t_L} - \partial_{t_R}) = i(\partial_t + \Omega_- \partial_\phi)/\kappa_- = i\zeta^-/\kappa_- \\
J &= H_0 - \bar{H}_0 = i(\partial_{t_R} + \partial_{t_L}) = i(\partial_t + \Omega_+ \partial_\phi)/\kappa_+ = i\zeta^+/\kappa_+.
\end{aligned}
\tag{B.6}
$$

From the above equations, we can see that $\partial_{t_L} + \partial_{t_R}$ and $\partial_{t_R} - \partial_{t_L}$ are the Killing vectors that vanish on the outer and inner horizons. Given the state dual to the scalar $\Phi = R(r)S(\theta)e^{-i\omega_L t_L - i\omega_R t_R}$, the eigenvalues of $H$ and $J$ are the monodromy parameters

$$
\begin{aligned}
H\Phi &= i(\partial_{t_L} - \partial_{t_R})\Phi = 2\alpha_- \Phi \\
J\Phi &= i(\partial_{t_R} + \partial_{t_L})\Phi = 2\alpha_+ \Phi.
\end{aligned}
\tag{B.7}
$$

This is again a direct consequence of the choice (B.1). For further discussion motivating (B.1), see [45]. From (B.6) and (B.7), we can also see that $2\alpha_\pm$ are the conserved charges associated with Wald's Killing vectors (B.2) that vanish on the inner and outer horizons.

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
