# Peer review of "Hidden Conformal Symmetries from Killing Towers with an Application to Large-D/CFT"

_SciPost Physics_

## Round 1 · Referee Report · Anonymous · 2022-3-27

Strengths

The manuscript is very well written. Computations are explicit and it is easy to follow. It summarises clearly prior results and embeds with existing literature.

Weaknesses

For the Kerr-AdS$_{D+1}$ BHs studied, it would be interesting to connect the hidden symmetries of the wave equation with the dual CFT$_{D}$. One expects the thermodynamic properties and greybody factors of the black hole to be tied to the dual CFT, so I was expecting more comments or discussion about how this ties in the more traditional context of AdS/CFT.

Report

This is an interesting manuscript that studies and unveils new connections between the analytic properties of the wave equation to the Killing tower. These new structures and corrections are important to the development of methods that would allow more control on dynamical properties of black holes, and therefore I find this manuscript valuable.

Requested changes

I think it might be of interest to the authors to cite https://arxiv.org/abs/2105.01069 which also explores hidden symmetries that are well defined for Schwarzschild. (There is also a subsequent paper https://arxiv.org/abs/2203.08832, but this one came out after this manuscript, so I don't expect it to be cited.)

---

## Round 1 · Referee Report · Anonymous · 2022-3-27

Strengths

1. The article connects two different techniques for studying hidden symmetries (Killing tensors and monodromy method).

2. The article identifies the algebra of conformal symmetries in the near-horizon limit of rotating black holes in arbitrary dimensions.

Weaknesses

None.

Report

This interesting article explores hidden conformal symmetries of Kerr black holes in arbitrary dimensions. Specifically, the authors combine two different methods associated with Killing tensors and monodromy, which have been used in different contexts in the past, to construct the algebra of hidden symmetries underlying the dynamics of scalar fields in Kerr geometries.

Over the last two decades, hidden symmetries of rotating black holes have been studied in two different contexts. Using the standard techniques of general relativity, separation of dynamical equations has been demonstrated in all dimensions, and the Killing tensors underlying such separation have been identified. Separately, in the context of AdS/CFT correspondence, new conformal symmetries have been found in the near-horizon limits of black holes in 4 and 5 dimensions. This article combines these two lines of research and uses the Killing tensors to identify the conformal symmetries in arbitrary dimensions and to analyze their algebra.

The article is very well written. It starts with a self-contained review of methods associated with Killing tensors and with monodromy used in studies of conformal symmetry. Then in sections 3 and 4 the authors combine these techniques to derive the algebraic structure of hidden conformal symmetry in arbitrary dimensions, and in section 5, the large D limit is analyzed. This article will be interesting to a wide audience of researchers working on general relativity and AdS/CFT correspondence, so I recommend it for publication.

I would suggest adding a couple of clarifications on page 19.

1. From the sentence "We want to propose a tensor equation T_{ab} = 0, which enforces the result (nabla T nabla Phi)." it is not clear what result is being enforced. If tensor T vanishes in a region, then all its derivatives vanish as well, so perhaps the last condition should read (nabla T nabla Phi)=0.

2. Before equation (3.42) the authors say "We thus propose the tensor equation." They should clarify whether this relation is derived or it is a new constraint imposed for convenience.

I leave implementation of these suggestions to authors' discretion, and I recommend the article for publication.

---

## Editorial Decision

resubmitted